# Eliciting Diverse Thinking Schemata for Large Reasoning Models

## Abstract

Large reasoning models (LRMs) have attracted increasing attention for their ability to solve complex mathematical problems by generating extended reasoning chains. In this work, we highlight a critical yet underexplored aspect of their reasoning process—thinking schemata, which we define as the distinct transitions between reasoning steps and the variety of solution paths the model produces. We observe a correlation between the diversity of thinking schemata and model performance, which motivates us to enhance diversity as a means to further improve reasoning potential and generalization ability. To this end, we propose **Di**verse **Sc**hemata Policy **O**ptimization (DiScO), a method to elicit diverse thinking schemata by first endowing the model with the capabilities to be aware of the thinking schemata in its reasoning chain and then encouraging their diversity through reinforcement learning. Experiments on multiple mathematical reasoning benchmarks demonstrate that DiScO consistently outperforms standard group relative policy optimization, with particularly pronounced gains on challenging datasets such as AIME, where our 7B and 32B DiScO models surpass the closed-source frontier LRMs by 2.9%-8.5%. Overall, our work suggests the important role that diversity of the reasoning procedure plays and points to scaling along the diversity dimension as a promising research direction.

## 1 Introduction

Large reasoning models (LRMs) have emerged as a promising paradigm for complex problem solving, where performance is achieved through generating multi-step chains of thought that decompose tasks into intermediate reasoning steps. Recent advances in scaling and reinforcement learning have further strengthened their ability to tackle challenging reasoning tasks, making them a central focus of research in artificial intelligence. Nevertheless, current approaches often impose overly rigid reasoning structures—such as strictly chain-like sequences (Wei et al., 2022; Zhang et al.) or explicitly organized trees and graphs (Yao et al., 2023; Long, 2023; Besta et al., 2024; Yao et al., 2024). While intuitive, these structures risk introducing strong priors that may not align with how human reasoning naturally unfolds, which is typically flexible, associative, and non-linear—more like a cloud of evolving perspectives than a fixed path.

To better capture this cognitive flexibility, we introduce the concept of *thinking schemata*[1]. Inspired by schema theory in cognitive neuroscience (Axelrod, 1973; Arbib, 1992; Fischbein & Grossman, 1997), we define thinking schemata as the distinct transitions (e.g., alternatively", on the other hand", etc.) between intermediate reasoning steps (i.e., *Reasoning Transitions*) and the variety of solution trajectories and outcomes that emerge from them (i.e., *Answer Candidates*), shown in Figure 1. Unlike previous views that regard reasoning as a linear deduction, our schemata definition centers on the dynamics of the reasoning process itself, abstracting away from rigid structures to instead measure the number of potential intermediate answers and the transitions that connect them. This abstraction encourages models to not only "think deep" but also "think diverse", promoting richer, more human-like reasoning trajectories.

---

[1] Compared to the previous parlance of schemata (Agarwal et al., 2025; Wen et al., 2025b; Chen et al., 2025; Shen et al., 2025) which mostly focus on the data with graph structure, the thinking schemata we defined in this work consider more regarding the thinking structure.

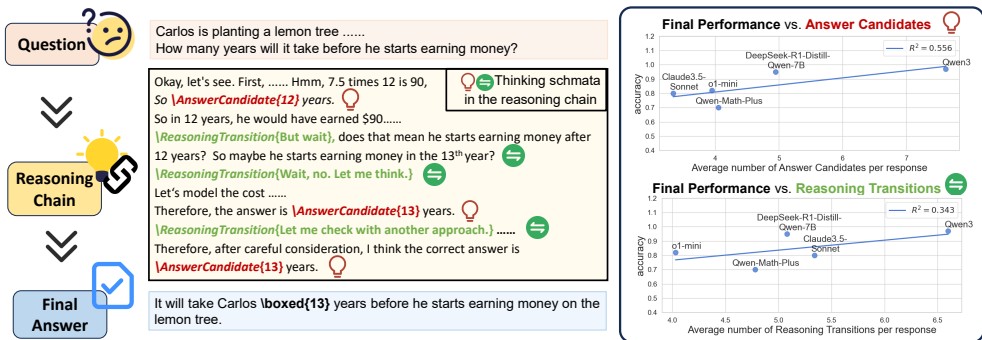

Figure 1: An illustration of the thinking schemata, including answer candidates and the reasoning transitions, that we defined and their relation with the final performance.

To concretize the above intuition, we further observe that the diversity of reasoning trajectories plays a crucial role in model performance, shown in Figure 1 and Section 2.2, and then introduce DiScO (**Di**verse **Sc**hemata Policy **O**ptimization), a framework designed to enhance thinking schemata during both training and inference. In training, we equip the model with a basic self-awareness of its reasoning process by having it annotate points of thought transition and enumerate possible intermediate answers. We then apply reinforcement learning to encourage the generation of diverse transitions and distinct solution paths, thereby strengthening its reasoning capacity. At inference time, we further promote diversity through two complementary strategies: (i) truncating the initial 20% of the reasoning chain to allow the model to "restart" from a fresh perspective, and (ii) filtering out duplicated reasoning by detecting the first point of repetition and removing the redundant span. Together, these mechanisms push the model to "think deep and think diverse," yielding more flexible and human-like reasoning trajectories.

We evaluate DiScO on a range of challenging mathematical reasoning benchmarks against both open-source and frontier reasoning models. Across settings, DiScO consistently outperforms comparable baselines at the 7B- and 32B-parameter scales, demonstrating substantial improvements in pass@1 accuracy. Notably, we also observe a clear relationship between the diversity of thinking schemata and model performance: models trained with DiScO generate more varied intermediate reasoning steps, which strongly aligns with gains in both accuracy and generalization. These results validate that promoting diverse thinking schemata is an effective and scalable way to strengthen reasoning ability, enabling DiScO not only to set new state-of-the-art results among open-source models but also to narrow the gap with leading proprietary systems. Furthermore, our findings suggest that scaling along the diversity dimension represents a promising and necessary direction for future research in advancing large reasoning models.

## 2 THINKING SCHEMATA

### 2.1 DEFINITIONS

*Schemata*, originally introduced in cognitive science (Casson, 1983; Rumelhart, 1984; Bartlett, 1995), refer to structured frameworks or mental templates that guide perception, understanding, and reasoning (Fischbein, 1999; Fischbein & Grossman, 1997; Cheng & Holyoak, 1985). Inspired by this theory, we define **Thinking Schemata** in the context of LRMs as the latent reasoning structure that governs how the model transitions between intermediate thoughts and arrives at potential solutions during multi-step problem solving.

Thinking schemata provide a higher-level abstraction of the reasoning dynamics within a model. Rather than focusing solely on final answers or local token-level behavior, we view thinking schemata as encompassing the broader cognitive pattern of the model's inference process. This includes how it shifts perspectives, explores solution space, and generates candidate reasoning paths.

*Diversity of Thinking Schemata:* In this work, we particularly focus on two key properties that characterize the diversity and richness of thinking schemata:

Table 1: Diversity results of sampled reasoning chains on the AMC 2023 benchmark.

| Model | Accuracy | Answer Candidates-avg | Reasoning Transition-avg |
|---|---|---|---|
| Qwen 3[2] | 97.5 | 7.62 | 6.60 |
| DeepSeek-R1-Distill-Qwen-7B (DeepSeek-AI, 2025) | 95.0 | 4.95 | 5.08 |
| o1-mini (Jaech et al., 2024) | 82.5 | 3.95 | 4.03 |
| Claude3.5-Sonnet (Anthropic, 2025) | 80.0 | 3.34 | 5.34 |
| Qwen-Math-Plus[2] | 70.0 | 4.05 | 4.78 |

**Reasoning Transition.** We define *Reasoning Transition* as the transition between semantically distinct reasoning steps or perspectives within a reasoning chain. These transitions reflect changes in reasoning mode—such as shifting from numerical computation to algebraic manipulation, or from a geometric intuition to a formal derivation. A model exhibiting more diverse Reasoning Transitions is considered to have greater cognitive flexibility and is better positioned to explore alternative reasoning routes.

**Answer Candidate.** The *Answer Candidate* refers to the set of distinct final or intermediate solutions the model proposes during its reasoning process. A model with diverse thinking schemata is expected to raise multiple plausible answers or sub-conclusions when reasoning, particularly in tasks with ambiguity, multi-path solutions, or when under uncertainty. The presence of multiple answer hypotheses is indicative of a broader inferential manifold being explored.

By capturing and promoting variation in both *Reasoning Transition* and *Answer Candidate*, our approach aims to elicit a richer distribution of thinking schemata, thus enhancing the reasoning capability and robustness of LRMs. An illustrative example is provided in Appendix E.2

## 2.2 DIVERSE THINKING SCHEMATA HELP LARGE REASONING MODELS

While LRMs have made substantial progress on complex tasks, we hypothesize that their performance can be further improved by increasing the diversity of their thinking schemata. Concretely, we expect that (1) a larger set of distinct answer candidates and (2) richer sequences of semantic transitions afford more opportunities for discovering correct solution trajectories, whereas (3) repetitive reasoning segments reflect a bias toward recycled templates and thus harm final performance.

**Experimental setup.** To evaluate these hypotheses, we sample multiple reasoning chains per input from a range of models (o1-mini (Jaech et al., 2024), DeepSeek-R1-Distill-Qwen-7B (DeepSeek-AI, 2025), Claude3.5-sonnet (Anthropic, 2025), Qwen-math-plus[2] and Qwen3[2]) on the AMC 2023 benchmark (Art of Problem Solving, 2023). Each sampled chain is annotated by Qwen3 using the labeling prompt described in Appendix C.2, producing per-instance metrics including the number of Answer Candidates ("answerCandidates-avg") and the number of semantic transitions ("reasoningTransition-avg"). Descriptive statistics for these metrics are reported in Table 1.

**Results.** We quantify the relationship between model performance (measured as accuracy) and diversity metrics by fitting simple linear regression models across multiple model–dataset points. Across models, we observe a consistent pattern: higher Answer Candidates-avg and more reasoning transitions are associated with improved performance. The coefficients of determination $R^2$ are shown in Figure 1.

**Discussion.** The empirical relationship between diversity metrics and accuracy supports the view that there is a potential that LRMs could benefit from exploring varied inferential trajectories rather than relying on a narrow set of cognitive templates. These findings highlight a gap in current approaches: although models can produce multiple reasoning chains, they rarely cultivate structured diversity in how solutions are pursued. To address this, we introduce a method that explicitly promotes diverse schemata in reasoning. By equipping models with mechanisms to recognize and regulate their own thinking schemata, and by reinforcing the generation of distinct transitions and solution paths, our approach aims to transform the observed correlations between diversity and performance into concrete improvements.

---

[2]https://bailian.console.aliyun.com/

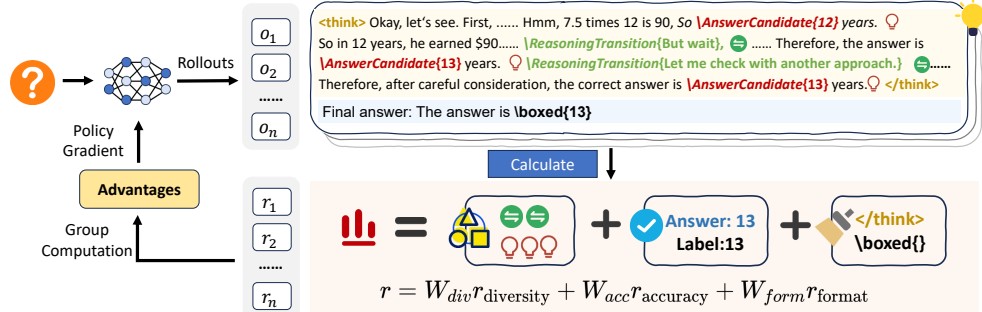

Figure 2: An illustration of the DiScO framework. During training, the model generates reasoning chains and receives rewards for the diversity of answer candidates and reasoning transitions, structured formatting, and accurate final outputs.

## 3 METHOD

We aim to enhance the reasoning ability of LRMs by explicitly modeling and encouraging diversity in their thinking schemata. Our approach consists of two key components. First, we equip the model with an annotation ability, enabling it to mark transitions in its reasoning process and identify possible intermediate answers, thereby fostering self-awareness of its own thought dynamics. Second, we introduce a reinforcement learning framework–Diverse Schemata Policy Optimization–that builds upon this annotation ability to encourage the generation of distinct reasoning transitions and diverse solution trajectories. Together, these components endow the model with the capacity to "think deep and think diverse", leading to more flexible and robust reasoning.

### 3.1 ANNOTATION ABILITY EQUIPMENT

We distill from Qwen-max (Qwen et al., 2024) to equip our models with the basic ability of self-awareness about the thinking schemata in its generated reasoning chain. We sampled 799 "question-reasoning-answer" triplets from the OpenR1-Math-220k (Hugging Face, 2025) dataset and have the Qwen-max to annotate the thinking schemata in the reasoning chains. The prompt we used for this task is shown in Appendix C.2 and the hyperparameters are shown in Section 4.1.

### 3.2 DIVERSE SCHEMATA POLICY OPTIMIZATION (DISCO)

To encourage large reasoning models to "think deep and think diverse," we adopt Group Relative Policy Optimization (GRPO) as the reinforcement learning backbone. GRPO (Shao et al., 2024; Guo et al., 2025) is a variant of PPO (Schulman et al., 2017), which avoids training an additional value function by leveraging groupwise relative comparisons of sampled outputs. Specifically, given a question $q$, GRPO samples a group of outputs $\{o_1, \ldots, o_G\}$ from the old policy $\pi_{\theta_{old}}$ and optimizes the policy $\pi_\theta$ with the following objective:

$$r_{i,t} = \frac{\pi_\theta(o_{i,t} \mid q, o_{i,<t})}{\pi_{\theta_{old}}(o_{i,t} \mid q, o_{i,<t})} \tag{1}$$

$$\mathcal{J}_{GRPO}(\theta) = \mathbb{E}_{[q, \{o_i\}]} \left[ \frac{1}{G} \sum_{i=1}^{G} \frac{1}{|o_i|} \sum_{t=1}^{|o_i|} \left\{ \min \left[ r_{i,t} \hat{A}_{i,t}, \text{clip}(r_{i,t}, 1-\epsilon, 1+\epsilon) \hat{A}_{i,t} \right] - \beta \mathbb{D}_{KL}[\pi_\theta \| \pi_{ref}] \right\} \right] \tag{2}$$

where $\hat{A}_{i,t}$ is the advantage estimated by comparing rewards across outputs in the same group, $\epsilon$ and $\beta$ are hyper-parameters, $\pi_{ref}$ is a frozen reference model.

**Reward Design.** Unlike standard GRPO, which typically relies on accuracy- and format-based rewards, we introduce a composite reward that explicitly encourages diverse schemata in reasoning:

$$R = W_{div} R_{\text{diversity}} + W_{acc} R_{\text{accuracy}} + W_{form} R_{\text{format}}, \tag{3}$$

where $W_*$ denotes the corresponding weight assigned to each reward component.

**Diversity Reward.** To capture richness in reasoning trajectories, we measure four complementary aspects:

$$R_{\text{diversity}} = S_{ansCnt} + S_{thoughtCnt} + S_{ansDiv} + S_{ansAcc}, \tag{4}$$

$$S_{ansCnt} = \min(N_{ans}, Max_{ansCnt}) \cdot w_{cnt}, \tag{5}$$

$$S_{thoughtCnt} = \min(N_{thought}, Max_{thoughtCnt}) \cdot w_{cnt}, \tag{6}$$

$$S_{ansDiv} = \min(N_{ans}^{uniq}, Max_{ansDiv}) \cdot w_{div}, \tag{7}$$

$$S_{ansAcc} = \min(N_{ans}^{true}, Max_{ansAcc}) \cdot w_{true}. \tag{8}$$

Here $N_{ans}$ counts answer candidates, $N_{thought}$ counts reasoning transitions, $N_{ans}^{uniq}$ measures unique solution candidates, and $N_{ans}^{true}$ counts correct intermediate candidates. $Max_*$ and $\omega_*$ are hyperparameters, where $Max_*$ denotes the maximum truncation limit for the corresponding $N_*$, and $\omega_*$ represents its associated weight.

The above terms directly reflect the two core aspects of thinking schemata: $N_{thought}$ captures the richness of **Reasoning Transitions**, while $N_{ans}$ and its variants ($N_{ans}^{uniq}$, $N_{ans}^{true}$) operationalize the diversity and reliability of **Answer Candidates**. By rewarding both dimensions jointly, the model is encouraged to explore alternative inferential routes while maintaining plausible and accurate solution hypotheses.

**Accuracy Reward.** The model is rewarded for producing a correct final answer:

$$R_{\text{accuracy}} = \mathbb{I}(\text{ans}) = \begin{cases} 1 & \text{if the final answer is correct,} \\ 0 & \text{otherwise.} \end{cases} \tag{9}$$

**Format Reward.** To ensure well-structured reasoning outputs (using the `</think>` tag to mark the end of thought and the `\boxed{}` to represent the final answer), we provide a lightweight reward for following the annotation format:

$$R_{\text{format}}(o) = 0.5 \cdot [\![\, \text{"</think>"} \in o \,]\!] + 0.5 \cdot [\![\, \text{"\boxed"} \in o \,]\!]. \tag{10}$$

By combining these components, DiScO aligns reinforcement learning not only with accuracy but also with the promotion of diverse and structured thinking schemata, bridging the empirical link between diversity and performance observed in our analysis.

### 3.3 Inference-Time Interventions for Enhancing Reasoning Diversity

To encourage deeper and more diverse reasoning during inference on complex tasks, we introduce two simple yet effective strategies that dynamically optimize the input reasoning chain to the model at inference time.

**Initial Truncation:** We allow the model to "forget" the initial 20% of its prior reasoning chain by truncating the beginning of the reasoning sequence and continuing generation. This strategy simulates a cognitive reset, giving the model more test-time "thinking space" to reframe the problem and potentially explore new reasoning trajectories.

**Truncation with Repetition Elimination:** In cases where the model exceeds the max generation length and produces highly repetitive reasoning—defined as the repetition of the same 15-word phrase multiple times—we apply a filtering step that removes duplicated segments. Specifically, we identify the first point of repetition and truncate the input to remove the duplicated span. This method prevents the model from being trapped in local loops and encourages it to explore more diverse and meaningful reasoning paths.

Together, these techniques aim to expand the inference-time search space, mitigate redundancy, and promote the generation of varied thinking schemata.

## 4 EXPERIMENTS

### 4.1 EXPERIMENT SETUP

**Datasets** We use DeepSeek-R1-Distill-Qwen-7B/32B as our base models and build our implementation on the VeRL framework (Sheng et al., 2024). During the supervised fine-tuning (SFT) stage, we leveraged a subset of the OpenR1-Math-220k dataset. The first 962 samples were annotated using the Qwen-Max[3] and Qwen3[3] APIs, with the annotation prompt provided in the Appendix C.2. After filtering, we retained 840 samples for training and 45 samples for validation. For the GRPO stage, we randomly sampled 8k instances from the DeepScaler (Shi et al., 2025) dataset within an accuracy range of 20%–80%, reserving 0.5% as the validation set. We use 8 rollouts per prompt, and a max length of 32768 tokens. Other training hyperparameters can be found in Appendix B.

To evaluate the model's reasoning capabilities, we select benchmarks of varying difficulty levels. For mathematical reasoning, we adopt GSM8K (Cobbe et al., 2021), MATH500 (Hendrycks et al., 2024), and the competition-level AIME 2024 (Art of Problem Solving, 2025a), AIME 2025 (Art of Problem Solving, 2025b), and AMC 2023 (Art of Problem Solving, 2023). For STEM reasoning, we use the challenging GPQA-Diamond (Rein et al., 2024) benchmark.

**Baselines** We leverage a diverse range of Frontier LLMs, Open-Sourced Reasoning LLMs as our baseline models. Frontier LLMs include leading proprietary models such as GPT-4o (Hurst et al., 2024), Claude 3.5-Sonnet, and the GPT-o1 series, which are recognized for their advanced reasoning capabilities. Open-Sourced Reasoning LLMs feature prominent models like DeepSeek-Coder-V2-Instruct (Zhu et al., 2024), Mathstral-7B (Mistral AI, 2024), NuminaMath-72B (Li et al., 2024), LLaMA3.1 series (Grattafiori et al., 2024), and Qwen2.5-Math-72B (Yang et al., 2024), which are widely benchmarked in mathematical tasks. The 7B and 32B parameter-scaled cohorts include base and instruction-tuned variants such as Qwen2.5-Math-7B/32B, ReasonFlux-7B/32B (Yang et al., 2025), and QwQ-32B (Qwen, 2024), which are evaluated to explore the trade-offs between model size and performance.

In the ablation study, we fine-tune DeepSeek-R1-Distill-Qwen-7B with standard SFT to obtain Qwen2.5-SFT-7B, which serves as the baseline for comparison with Qwen2.5-Anno-7B ( with annotation capability). We further performe GRPO training without the diversity reward on Qwen2.5-Anno-7B and Qwen2.5-Anno-32B, obtaining Qwen2.5-Anno-GRPO-7B and Qwen2.5-Anno-GRPO-32B, which are compared with our DiScO models.

### 4.2 RESULTS

**Main results** Table 2 reports Pass@1 accuracy across multiple mathematical reasoning benchmarks. Among frontier LLMs, GPT-o1-mini and GPT-o1-preview perform strongly but show limited accuracy on competition datasets such as AIME. For open-source models, math-specialized systems like Qwen2.5-Math-72B and DeepSeek-V3 outperform earlier instruction-tuned baselines, while ReasonFlux-32B sets a strong baseline with 91.2% on MATH-500 and 56.7% on AIME 2024.

Our proposed DiScO models achieve the best results at both scales. DiScO-7B obtains 95.4% on MATH-500, 86.7% on AIME 2025, and an average of 78.1%, clearly surpassing other 7B models. DiScO-32B further improves to 93.8% on MATH-500, 86.7% on AIME 2025, 66.7% on AIME 2024, and 83.2% average, outperforming both frontier and open-source baselines.

These results demonstrate that DiScO substantially advances mathematical reasoning performance, particularly on challenging competition-level tasks. Conversely, on datasets where baselines already perform well, the gains are more modest. Taken together, this pattern suggests that promoting thinking schemata diversity is especially valuable when models face problems that require extensive exploration of the reasoning space.

**Ablation results for diverse reward design** Table 3 presents Pass@1 accuracy across six mathematical reasoning benchmarks, comparing the full DiScO model with ablated variants and the Qwen2.5-Anno-GRPO-7B baseline.

---

[3] https://bailian.console.aliyun.com/

Table 2: Pass@1 accuracy comparison on various reasoning benchmarks.

| Model | MATH-500 | AIME 2024 | AIME 2025 | AMC 2023 | GPQA-Diamond | GSM8K | Average |
|---|---|---|---|---|---|---|---|
| **Frontier LLMs** | | | | | | | |
| GPT-4o | 76.6 | 16.7 | 26.7 | 47.5 | 53.6 | 89.5 | 51.7 |
| Claude3.5-Sonnet | 78.3 | 16.0 | 43.3 | 87.5 | 40.4 | 96.4 | 60.3 |
| GPT-o1-preview | 85.5 | 44.6 | 46.7 | 90.0 | **73.3** | 94.9 | 72.5 |
| GPT-o1-mini | 90.0 | 56.7 | 50.8 | 95.0 | 60.0 | 95.8 | 74.7 |
| **Open-Sourced Reasoning LLMs** | | | | | | | |
| DeepSeek-Coder-V2-Instruct | 75.3 | 13.3 | - | 57.5 | 44.3 | 94.9 | 57.1 |
| Mathstral-7B-v0.1 | 57.8 | 0.0 | 0.0 | 37.5 | 9.1 | 77.1 | 30.3 |
| NuminaMath-72B-CoT | 64.0 | 3.3 | - | 70.0 | 35.3 | 91.4 | 52.8 |
| LLaMA3.1-8B-Instruct | 51.4 | 6.7 | 0.0 | 25.0 | 30.4 | 82.4 | 32.7 |
| LLaMA3.1-70B-Instruct | 65.4 | 16.7 | 3.3 | 50.0 | 48.0 | 91.7 | 45.9 |
| LLaMA3.1-405B-Instruct | 73.8 | 23.3 | 10.0 | 50.0 | 49.0 | **96.8** | 50.5 |
| Qwen2.5-Math-72B-Instruct | 85.6 | 30.0 | 26.7 | 70.0 | 42.9 | 95.5 | 58.5 |
| DeepSeek-V3 | 90.2 | 47.7 | 39.2 | 80.0 | 59.1 | 94.2 | 68.4 |
| ReasonFlux-32B | 91.2 | 56.7 | 37.2 | 85.0 | 61.2 | 79.3 | 68.4 |
| rStar-Math | 89.4 | 50.0 | - | 87.5 | - | 95.0 | 80.5 |
| **DiScO-32B** | **93.8** | **86.7** | **66.7** | **97.5** | 58.1 | 96.4 | **83.2** |
| **7B-Level Base Model** | | | | | | | |
| Qwen2.5-Math-7B | 58.8 | 16.7 | 3.3 | 22.5 | 28.3 | **95.0** | 37.4 |
| SuperCorrect-7B | 70.2 | 26.7 | 13.3 | 37.5 | 20.7 | 84.7 | 42.2 |
| DeepSeek-R1-Distill-Qwen-7B | 64.0 | 36.7 | 13.3 | 95.0 | 10.6 | 70.2 | 48.3 |
| Qwen2.5-Math-7B-Instruct | 82.6 | 13.3 | 16.7 | 28.3 | **62.5** | **95.0** | 49.7 |
| ReasonFlux-7B | 88.6 | 36.7 | 36.7 | 80.0 | 35.9 | 83.9 | 60.3 |
| **DiScO-7B** | **95.4** | **83.3** | **53.3** | **95.0** | 46.0 | 92.5 | **77.6** |
| **32B-Level Base Model** | | | | | | | |
| Qwen2.5-32B-Instruct | 79.4 | 16.5 | 13.3 | 64.0 | 49.5 | 94.4 | 52.9 |
| Sky-T1-32B-preview | 89.5 | 43.3 | 36.7 | - | 56.8 | 94.8 | 64.2 |
| ReasonFlux-32B | 91.2 | 56.7 | 37.2 | 85.0 | 61.2 | 79.3 | 68.4 |
| QwQ-32B-preview | 90.6 | 50.0 | 46.7 | 75.0 | **65.2** | 91.2 | 69.8 |
| DeepSeek-R1-Distill-Qwen-32B | 93.4 | 80.0 | 60.0 | 97.5 | 63.6 | 94.0 | 81.4 |
| **DiScO-32B** | **93.8** | **86.7** | **66.7** | **97.5** | 58.1 | **96.4** | **83.2** |

Table 3: Ablation results of diverse reward designs, showing Pass@1 accuracy across benchmarks.

| Model | MATH-500 | AIME 2024 | AIME 2025 | AMC 2023 | GPQA-Diamond | GSM8K | Average |
|---|---|---|---|---|---|---|---|
| DiScO-7B | **94.8** | **83.3** | **50.0** | **95.0** | 43.4 | 92.3 | **77.0** |
| −AnswerCandidate | 93.0 | 76.7 | **50.0** | 92.5 | 49.5 | 93.3 | 75.8 (-1.2) |
| −ReasoningTransition | 93.6 | 80.0 | 46.7 | 92.5 | **51.0** | 93.3 | 76.2 (-0.8) |
| Qwen2.5-Anno-GRPO-7B | 92.6 | 56.7 | **50.0** | 90.0 | 49.5 | **93.6** | 72.1 (-4.9) |

We observe that both dimensions of schemata diversity—*Answer Candidate* and *Reasoning Transition*—contribute substantially to performance. Removing either reward leads to consistent drops in average accuracy: −AnswerCandidate decreases from 77.0% to 75.8%, while −ReasoningTransition reduces it to 76.2%. This indicates that the two objectives capture complementary aspects of reasoning diversity: Answer Candidate rewards encourage exploration of multiple solution hypotheses, whereas Reasoning Transition rewards foster richer transitions between intermediate reasoning steps.

**Ablation results for inference** Figure 3 summarizes the average Pass@1 accuracy across six mathematical reasoning benchmarks. Detailed results can be found in Appendix D. At the 7B scale, truncation consistently improves baseline models: for example, DeepSeek-R1-Distill-Qwen-7B rises from 48.3% to 56.2% with repetition elimination. Among all 7B variants, DiScO-7B achieves the highest accuracy (77.6%) when combined with initial truncation.

At the 32B scale, truncation further amplifies performance. DeepSeek-R1-Distill-Qwen-32B improves steadily from 81.4% to 81.6%, while Qwen2.5-Anno-32B and Qwen2.5-Anno-GRPO-32B also show moderate gains. The best overall results come from DiScO-32B, which reaches 83.2% with repetition elimination, establishing the strongest performance across all models.

Overall, the results show that truncation strategies yield stable gains across scales and benchmarks, with the largest benefits observed on challenging datasets such as GPQA-Diamond and AIME 2025. These findings further confirm that our lightweight truncation methods effectively reduce

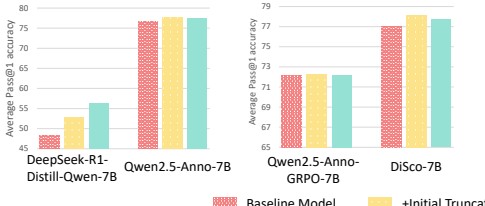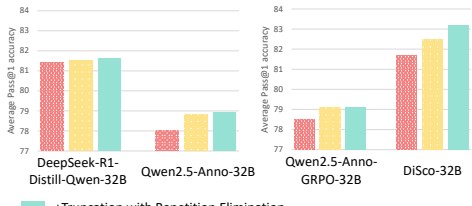

Figure 3: Ablation results for three inference strategies, where each bar denotes the average Pass@1 accuracy across six mathematical reasoning benchmarks (MATH-500, AIME 2024, AIME 2025, AMC 2023, GPQA-Diamond, and GSM8K).

redundancy in reasoning and improve overall accuracy, particularly when combined with diversity-oriented training on larger scales.

### 4.3 ANALYSIS

**Thinking Schemata Analysis** We analyze the diversity of reasoning chains generated by baseline models (Qwen2.5-Anno-GRPO-7B) and our diversity-enhanced variants (DiScO-7B) across multiple datasets (Table 4). Compared to the baseline, DiScO-7B consistently increases both the average number of reasoning transitions (#RT-avg) and answer candidates (#AC-avg), indicating richer exploration of the solution space. However, it is crucial to note that these metrics are not monotonically beneficial, but quality matters more than quantity. For instance, DeepSeek-R1-Distill-Qwen-7B generates high #RT-avg and #AC-avg values on certain datasets, such as AIME 2025, but these inflated numbers stem from repetitive content generation rather than meaningful exploration, as detailed in Appendix E.1. The true-answer ratio (TAR) which measures the proportion of correct answers among all generated candidates, serves as a key indicator of robustness, i.e. model's ability to consistently identify correct solutions when exploring diverse reasoning paths. Notably, DiScO-7B improves TAR compared to baselines in 5 out of 6 datasets, demonstrating that greater schemata diversity not only enriches reasoning dynamics but also enhances the robustness. Our analysis reveals that meaningful diversity, characterized by reduced repetitive content (RC) and increased unique answers (UA), is more important than raw counts of transitions or candidates. DiScO successfully reduces homogenization in reasoning where it maintains lower repetitive content while generating more unique intermediate perspectives. This balance between exploration and focus enables the model to avoid both the trap of repetitive reasoning loops and the inefficiency of unfocused exploration. Overall, these findings confirm that encouraging controlled variation in both transitions and candidate answers expands the model's effective reasoning space, leading to more robust and flexible problem-solving behavior, particularly on challenging mathematical reasoning tasks.

Table 4: Diversity measurement of sampled reasoning chains. "GSM8k-100", "GPQA-100", and "MATH-100" denote subsets of 100 samples randomly drawn from GSM8k, GPQA-Diamond, and MATH 500, respectively. "Distill-7B" and "GRPO-7B" refer to DeepSeek-R1-Distill-Qwen-7B and DeepSeek-R1-GRPO-Qwen-7B, respectively. Here, #RT-avg = average number of Reasoning Transitions, #AC-avg = average number of Answer Candidates, RC = Repetitive Content, UA = Unique Answer, and TAR = True-Answer Ratio.

| Dataset | Model | #RT-avg ↑ | #AC-avg ↑ | RC ↓ | UA↑ | TAR↑ |
|---------|-------|-----------|-----------|------|------|------|
| AIME 2024 | Distill-7B | 5.73 | 4.07 | **0.80** | 2.33 | 0.48 |
| | GRPO-7B | **12.86** | **15.21** | 2.76 | **7.07** | 0.23 |
| | DiScO-7B | 12.24 | 8.34 | 1.31 | 4.45 | **0.51** |
| AIME 2025 | Distill-7B | **57.76** | **22.00** | 2.21 | **6.28** | 0.08 |
| | GRPO-7B | 9.96 | 9.11 | **1.36** | 4.89 | 0.26 |
| | DiScO-7B | 9.76 | 8.62 | 1.52 | 5.00 | **0.27** |
| AMC 2023 | Distill-7B | **8.72** | 7.80 | 1.57 | 2.42 | **0.65** |
| | GRPO-7B | 5.95 | 6.90 | **1.45** | 3.55 | 0.45 |
| | DiScO-7B | 8.64 | **11.26** | 1.97 | **4.36** | 0.50 |
| GSM8k-100 | Distill-7B | 4.79 | 6.44 | **1.44** | 2.40 | 0.41 |
| | GRPO-7B | 6.13 | 8.52 | 1.87 | **3.67** | 0.45 |
| | DiScO-7B | **6.32** | **10.26** | 1.93 | 3.35 | **0.50** |
| GPQA-100 | Distill-7B | 2.68 | **18.21** | 1.96 | **4.22** | 0.02 |
| | GRPO-7B | 10.99 | 6.01 | **1.10** | 3.82 | 0.06 |
| | DiScO-7B | **14.59** | 6.92 | 1.45 | 4.02 | **0.08** |
| MATH-100 | Distill-7B | **14.86** | **10.56** | 1.69 | 2.73 | 0.43 |
| | GRPO-7B | 7.98 | 6.64 | 1.38 | **2.79** | **0.47** |
| | DiScO-7B | 11.64 | 7.05 | **1.33** | 2.55 | 0.46 |

## 5 DISCUSSION

**Diversity-based scaling** An important implication of our work is that scaling reasoning diversity itself could be a promising direction. Encouraging diverse thinking schemata through reinforcement

learning requires several times more computation at the same parameter scale, but the improvements on challenging benchmarks justify this cost. Similar to recent findings that LRMs increasingly rely on compute-intensive post-training, such as RL and reward modeling (Ji et al., 2025), we argue that diversity-based scaling is a worthwhile yet demanding direction, calling for both greater resources and more efficient optimization strategies, making it a crucial and challenging path for the next generation of reasoning models.

## 6 RELATED WORK

### 6.1 LARGE REASONING MODELS (LRMs)

Recent advances in Large Reasoning Models, such as OpenAI o1 and DeepSeek-R1, highlight the central role of reinforcement learning (RL) in enabling reasoning capabilities that pre-training alone cannot provide, with especially notable gains in mathematical reasoning (He et al., 2025) and code generation (Zhuo et al., 2024). DeepSeek-R1, for instance, demonstrates that large-scale RL with structured accuracy or test-based rewards, implemented via Group Relative Policy Optimization (GRPO), can induce sophisticated reasoning behaviors even before downstream alignment. As RL becomes a standard mechanism for improving LRMs, recent work has increasingly emphasized not only correctness but also diversity in reasoning, inspired by Yao et al. (2025). Consequently, a growing line of research investigates RL methods that explicitly encourage diverse multi-step reasoning. Recently, some work focuses on improving GRPO via more effective reward design. Zhang & Zuo (2025) introduces enhancements to GRPO for mathematical reasoning, incorporating a length-dependent accuracy reward, explicit penalties for incorrect answers, and a difficulty-aware advantage reweighting strategy, collectively improving learning efficiency. Meanwhile, Hu et al. (2020) study how to adaptively leverage a shaping reward by formulating its use as a bi-level optimization problem. Given GRPO's widespread use and demonstrated effectiveness across LRM training pipelines, we also build on this framework in our method.

### 6.2 REASONING PATTERNS IN LARGE REASONING MODELS

Researchers have investigated reasoning patterns in LRMs and their effects on solving math problems. Minegishi et al. (2025) introduces "Topology of Reasoning," where hidden-state clusters at each reasoning step form structures whose cycle frequency, graph diameter, and small-world characteristics correlate with model capacity and task difficulty, offering interpretable graph-theoretic insights into why reasoning-optimized LLMs perform better. Tian et al. (2025) presents a multi-round test-time thinking strategy where a model, given only its previous final answer (not its chain of thought), re-answers the same question across rounds—yielding consistent accuracy improvements without extra training. Marjanović et al. (2025) reveals that LRMs construct structured, multi-stage chains—starting with problem definition, followed by iterative "blooming" breakdowns and "reconstruction" reflections, before a final decision—showing that overly long reasoning degrades performance. An et al. (2025) argues LRMs often bloat reasoning steps—"overthinking"—and proposes dynamically pruning inefficient sub-patterns in multi-stage chains, yielding more concise, resource-efficient, and accurate outcomes. (Yang et al., 2025) enhances LRMs by guiding them through efficient multi-step reasoning with scalable, hierarchical thought templates, outperforming flat chain-of-thought methods. Wen et al. (2025a) shows that while smaller LLMs benefit from structured "thinking" patterns like decomposition, self-ask, self-debate, and self-critic, larger models perform best with simpler, unstructured monologue-style reasoning. Lee et al. (2025) proposes automatically extracting, clustering, and interpreting diverse chain-of-thought strategies from outputs to predict and steer LRMs toward more effective patterns.

Our definition of schemata differs from, and is more general than, the aforementioned thinking patterns, as it does not rely on any manual attribution or classification of reasoning chains. Instead, we focuses on the answer candidates and the transitions between reasoning trajectories, aiming to improve the performance of LRMs by encouraging the diverse generation of these thinking schemata.

## 6.3 DIVERSITY IN LARGE REASONING MODELS

To improve the training performance of LRMs, recent studies have proposed methods to explored diversity in LRMs during training. Yao et al. (2025) investigate the importance of promoting diversity during RL training and introduce Potential@k, a metric quantifying an LLM's reasoning potential after RL training. Their work demonstrates a strong correlation between solution diversity and performance. To leverage this, the method integrates a token-level diversity objective into R1-zero training, enhancing exploration while maintaining stability. Wang et al. (2025) focus on local branching by identifying high-entropy tokens as key divergence points and optimizing RL updates around them. Token-level methods, however, operate on localized entropy signals and therefore encourage diversity only at the micro-level. In contrast, an emerging line of trajectory-level approaches seeks to promote diversity at the scale of entire reasoning paths. One prominent direction draws on GFlowNets, which sample structured objects, such as reasoning traces, with probability proportional to reward, thereby enabling diverse and globally consistent exploration. Hu et al. (2023) use Sub-trajectory Balance to amortize posterior sampling in LLMs. Yu et al. and Nair et al. (2025) formulate multi-step reasoning as flows on Directed Acyclic Graphs(DAGs) to encourage varied reasoning paths under minimal supervision; and Younsi et al. (2025) trains a PRM from MCTS data and fine-tunes a step-level GFlowNet with Subtrajectory Balance and PRM-based multiplicative rewards to generate accurate, diverse trajectories. Despite these successes, trajectory-level GFlowNet methods typically rely on structural assumptions such as a DAG-based formulation of trajectories and the intrinsic flow-matching constraints required by the framework. These inductive biases guide exploration but can constrain the reasoning space, limiting the model's ability to capture free-form, spontaneous reasoning patterns that do not align with the assumed structure.

While token-level methods improve local branching behavior, their reliance on localized entropy signals restricts diversity to micro-level perturbations rather than capturing meaningful variations in reasoning strategy. GFlowNet-based approaches operate at the trajectory level but primarily optimize the probability of generating high-reward trajectories, achieving "reward-proportional" diversity rather than truly semantic diversity. In contrast, our method directly evaluates and shapes complete rollout trajectories, focusing on how they shift perspectives, explore alternative solution avenues, and form globally distinct reasoning paths. This trajectory-level view captures richer forms of exploratory reasoning and leads to genuine semantic diversity that better reflects the breadth of human-like problem-solving.

## 7 CONCLUSION

In this work, we introduce the notion of thinking schemata as a new perspective for understanding and improving the reasoning process of large reasoning models. We show that existing models tend to follow narrow and repetitive schemata, limiting their ability to generalize and explore alternative reasoning paths. To address this, we proposed DiScO, a reinforcement learning framework that equips models with self-awareness of their reasoning dynamics and explicitly promotes diversity in transitions and solution trajectories. Extensive experiments across multiple mathematical reasoning benchmarks demonstrated that DiScO substantially improves both accuracy and flexibility, narrowing the gap with frontier models. We believe that encouraging diverse thinking schemata offers a promising direction for developing more robust and human-like reasoning systems.

ETHICS STATEMENT

This work focuses on analyzing the working mechanisms of large reasoning models, as such, it does not present any increased risks of harm beyond the existing norms of natural language processing or computational linguistics research. The associated risks include using a model trained on vast amounts of text, which may inadvertently contain biases. Another concern is the potential misuse of the model for generating misleading or harmful content. However, such a scenario is unlikely in our work, as we concentrate on reasoning tasks with fixed answers.

REPRODUCIBILITY STATEMENT

To ensure the reproducibility of our work, we have made several efforts that are documented throughout this paper. Our experiments utilize the open-source models described in Section 4.1. The prompts used in our experiments are detailed in Appendix C. The complete code for our implementation, including inference processes, is provided in the supplementary materials. All datasets used in our experiments are described comprehensively in Section 4.1, and the supplementary code includes all data processing steps and any preprocessing applied. We encourage other researchers to consult these references for replicating our findings.

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

## A  THE USE OF LARGE LANGUAGE MODELS

We disclose that Large Language Models(LLMs) were utilized solely for our paper writing, including grammar correction and wording refinement. LLMs were not used to generate research ideas or perform analyzes.

## B  HYPERPARAMETERS

During the SFT stage, we adopt the LLaMA-Factory framework (Zheng et al., 2024) with a context length of 32K, learning rate of 1e-5, and train each model for 3 epochs. For reinforcement learning, we employ the VeRL framework (Sheng et al., 2024) with a 16K context length, 8 rollouts, and batch sizes of 256 for the 7B model and 512 for the 32B model. The 7B model is trained for 4 epochs (112 steps) on 8 nodes with 8×H20 GPUs, requiring approximately 55 hours, while the 32B model is trained for 4 epochs (72 steps) on 16 nodes with 8×H20 GPUs, taking about 36 hours. We use checkpoints at step 50 (7B) and step 40 (32B), where both models achieve peak accuracy. Hyperparameter details are summarized in Table 5. During inference, we set temperature to 0.0 and top_k to 1.0, applying greedy decoding with vLLM (Kwon et al., 2023). The prompts used for training and inference are demonstrated in Appendix C.3 and Appendix C.1, respectively.

| Hyperparameter | Value |
| --- | --- |
| $\beta$ | 0.001 |
| $\epsilon$ | 0.2 |
| $W_{acc}$ | 2.0 |
| $W_{div}$ | 1.0 |
| $W_{form}$ | 1.0 |
| $\omega_{wnt}$ | 0.1 |
| $\omega_{div}$ | 0.2 |
| $\omega_{true}$ | 0.3 |
| $Max_{ansCnt}$ | 15 |
| $Max_{thoughtCnt}$ | 20 |
| $Max_{ansDiv}$ | 15 |
| $Max_{ansAcc}$ | 15 |

Table 5: Hyperparameters used in GRPO training.

## C  PROMPT DESIGN

### C.1  INFERENCE PROMPT DESIGN

The adopted prompt for inference is shown below. To ensure that the model engages in thorough reasoning, we follow the recommendation to enforce the model to initiate its response with "`<think>\n`" at the beginning of every output[4].

> **Inference prompt design**
>
> Try to solve the following question step by step. If the final answer is obtained, use \\boxed{}
> to represent it.
> ### Question:{question}
> `<|Assistant|><think>\n`

### C.2  LABELING PROMPT DESIGN

As mentioned in Section 2.2, we sample multiple reasoning chains from a range of models on the AMC 2023 benchmark, with each chain annotated by Qwen3. The adopted labeling prompt is shown

---

[4]`https://huggingface.co/deepseek-ai/DeepSeek-R1-Distill-Qwen-7B`

below. In the code implementation, we use the two tags "possibleAnswer" and "thoughtchange" to represent "Answer Candidate" and "Reasoning Transition" respectively.

---

**Labeling prompt design**

The following is the reasoning chain that is used to answer a difficult math problem. Please process the reasoning chain according to the following rules:

1. Label all segments that are potentially final results in the reasoning chain with \\possibleAnswer{} format. DO NOT label all the possible intermediate results, ONLY label the ones that could be the final answers, no matter it's correct or wrong. Label as many as you could.

2. An example of the \\possibleAnswer{} annotation: "Wait, 5 times 360 is 1800, and 1800 divided by 36. Let's do that division: 1800 ÷ 36. Hmm, 36 times 50 is 1800, right? Because 36 x 50 is 1800. So, 1800 ÷ 36 = \\possibleAnswer{50}. Therefore, the degrees for cherry pie would be \\possibleAnswer{50} degrees."

3. Label all segments that indicate a shift in reasoning within the text reasoning chain using the \\thoughtchange{} format. Label as many as you could.

4. An example of the \\thoughtchange{} annotation: "\\thoughtchange{Wait, maybe} I messed up the dailyprogress.\\thoughtchange{Wait, hold on}. If the original totaltime is T days, then when they switch to the newequipment after 1/3 of the tunnel is done, whichtook T/3 days, and then the remaining 2/3 is doneat a slower daily rate"

5. DO NOT change other parts and keep them exactly the same as the the original solution.

### original solution:
{solution}
### Result:

---

## C.3 TRAINING PROMPT DESIGN

The adopted prompt for SFT and GRPO training is shown below. In the code implementation, we use the two tags "possibleAnswer" and "thoughtchange" to represent "Answer Candidate" and "Reasoning Transition" respectively.

---

**Training prompt design**

Try to solve the following question step by step. Please show your reasoning chain according to the following rules:

1. First thinks about the reasoning chain in the mind and then provides the user with the answer. The reasoning chain is enclosed within `<think>` `</think>` tags, i.e., `<think>` reasoning chain here `</think>`. If the final answer is obtained, use \\boxed{} to represent it.

2. Label all segments that are potentially final results in the reasoning chain with \\possibleAnswer format. DO NOT label all the possible intermediate results, ONLY label the ones that could be the final answers, no matter it's correct or wrong. Label as many as you could.

3. An example of the \\possibleAnswer annotation: "Wait, 5 times 360 is 1800, and 1800 divided by 36. Let's do that division: 1800 ÷ 36. Hmm, 36 times 50 is 1800, right? Because 36 x 50 is 1800. So, 1800 ÷ 36 = \\possibleAnswer50. Therefore, the degrees for cherry pie would be \\possibleAnswer50 degrees."

4. Label all segments that indicate a shift in reasoning within the text reasoning chain using the \\thoughtchange format. Label as many as you could.

5. An example of the \\thoughtchange annotation: "\\thoughtchangeWait, maybe I messed up the dailyprogress. \\thoughtchangeWait, hold on. If the original totaltime is T days, then when they switch to the newequipment after 1/3 of the tunnel is done, which took T/3 days, and then the remaining 2/3 is doneat a slower daily rate."

### Question:{question}
`<|Assistant|><think>`

---

# D ABLATION RESULTS FOR INFERENCE

As mentioned in Section 4.2, Table 6 demonstrates detailed ablation results for two inference strategies. Overall, the results show that truncation strategies yield stable gains across scales and benchmarks. In particular, DiScO-7B improves from 50.0% to 53.3% on AIME 2025 with truncation, while DiScO-32B achieves the most pronounced gain on GPQA-Diamond, rising from 53.5% to 58.1% with truncation combined with repetition elimination. These findings further confirm that our lightweight methods effectively reduce redundancy in reasoning and improve overall accuracy.

Table 6: Pass@1 accuracy comparison on various mathematical reasoning benchmarks.

| Model | MATH-500 | AIME 2024 | AIME 2025 | AMC 2023 | GPQA-Diamond | GSM8K | Average |
|---|---|---|---|---|---|---|---|
| Qwen-plus-latest | 85.0 | 46.7 | 26.7 | 97.5 | 8.1 | 94.3 | 59.7 |
| **7B-Level Base Model** | | | | | | | |
| DeepSeek-R1-Distill-Qwen-7B | 64.0 | 36.7 | 13.3 | 95.0 | 10.6 | 70.2 | 48.3 |
| +Initial Truncation | 78.8 | 36.7 | 20.0 | 95.0 | 13.6 | 72.5 | 52.8 |
| +Truncation with Repetition Elimination | 81.8 | 43.3 | 26.7 | 95.0 | 17.2 | 72.9 | 56.2 |
| Qwen2.5-SFT-7B | 93.0 | 63.3 | 40.0 | 92.5 | 48.0 | 91.0 | 71.3 |
| +Initial Truncation | 93.0 | 63.3 | 40.0 | 92.5 | 48.0 | 92.9 | 71.6 |
| +Truncation with Repetition Elimination | 93.2 | 66.7 | 40.0 | 92.5 | 50.0 | 92.7 | 72.5 |
| Qwen2.5-Anno-7B | 93.6 | **83.3** | 46.7 | 95.0 | 48.0 | 93.5 | 76.7 |
| +Initial Truncation | 93.6 | **83.3** | 46.7 | **97.5** | 51.0 | **93.7** | **77.6** |
| +Truncation with Repetition Elimination | 93.6 | **83.3** | 46.7 | **97.5** | 49.5 | **93.7** | 77.4 |
| Qwen2.5-Anno-GRPO-7B | 92.6 | 56.7 | 50.0 | 90.0 | 49.5 | 93.6 | 72.1 |
| +Initial Truncation | 92.8 | 56.7 | 50.0 | 90.0 | 50.0 | 93.6 | 72.2 |
| +Truncation with Repetition Elimination | 92.8 | 56.7 | 50.0 | 90.0 | 50.0 | 93.6 | 72.2 |
| DiScO-7B | 94.8 | **83.3** | 50.0 | 95.0 | 43.4 | 92.3 | 76.5 |
| +Initial Truncation | 95.4 | **83.3** | **53.3** | 95.0 | 46.0 | 92.5 | **77.6** |
| +Truncation with Repetition Elimination | **95.6** | **83.3** | 50.0 | 95.0 | 46.0 | 92.7 | 77.1 |
| **32B-Level Base Model** | | | | | | | |
| DeepSeek-R1-Distill-Qwen-32B | 93.4 | 80.0 | 60.0 | **97.5** | 63.6 | 94.0 | 81.4 |
| +Initial Truncation | 93.6 | 80.0 | 60.0 | **97.5** | **64.1** | 94.0 | 81.5 |
| +Truncation with Repetition Elimination | **93.8** | 80.0 | 60.0 | **97.5** | **64.1** | 94.0 | 81.6 |
| Qwen2.5-Anno-32B | 92.2 | 73.3 | 63.3 | 95.0 | 49.0 | 95.3 | 78.0 |
| +Initial Truncation | 92.2 | 73.3 | 63.3 | 95.0 | 53.5 | 95.3 | 78.8 |
| +Truncation with Repetition Elimination | 92.2 | 73.3 | 63.3 | 95.0 | 54.0 | 95.5 | 78.9 |
| Qwen2.5-Anno-GRPO-32B | 92.8 | 76.7 | 46.7 | **97.5** | 61.1 | 96.0 | 78.5 |
| +Initial Truncation | 93.6 | 76.7 | 46.7 | **97.5** | **64.1** | 96.0 | 79.1 |
| +Truncation with Repetition Elimination | **93.8** | 76.7 | 46.7 | **97.5** | **64.1** | 96.0 | 79.1 |
| DiScO-32b | 91.4 | **86.7** | **66.7** | **97.5** | 53.5 | 94.5 | 81.7 |
| +Initial Truncation | 92.0 | **86.7** | **66.7** | **97.5** | 57.1 | 94.8 | 82.5 |
| +Truncation with Repetition Elimination | **93.8** | **86.7** | **66.7** | **97.5** | 58.1 | **96.4** | **83.2** |

# E CASE STUDY

## E.1 EXAMPLE OF REPETITIVE GENERATION

We observe that DeepSeek-R1-Distill-Qwen-7B tends to recycle its prior reasoning and generate highly repetitive content, especially on challenging problems.

---

**Example: Response generated by DeepSeek-R1-Distill-Qwen-7B**

Okay, so I'm trying to figure out this organic chemistry problem. The starting material is (2,2-dimethylbut-3-en-1-yl)oxybenzene, ....... So the structure is: CH2-CH(CH3)-CH2-CH2. But with a double bond between carbons 3 and 4, so it's actually: CH2-CH(CH3)-CH=CH2\n

Wait, no, that's not correct. The butenyl group is but-3-en-1-yl, so the structure is: CH2-CH(CH3)-CH2-CH2. But with a double bond between carbons 3 and 4, so it's:CH2-CH(CH3)-CH=CH2.

Wait, no, that's not correct. The butenyl group is but-3-en-1-yl, so the structure is: CH2-CH(CH3)-CH2-CH. But with a double bond between carbons 3 and 4, so it's: CH2-CH(CH3)-CH=CH2.

Wait, no, that's not correct. The butenyl group is but-3-en-1-yl, so the structure is:CH2-CH(CH3)-CH2-CH2. But with a double bond between carbons 3 and 4, so it's: CH2-CH(CH3)-CH=CH2.

......

---

## E.2 EXAMPLE OF THINKING SCHEMATA

Figure 4 presents an example that demonstrates the thinking schemata within a reasoning chain.

Specifically, *ReasoningTransition* reflects changes such as shifting from setting up equations to algebraic manipulation, or from a numerical computing to a formal derivation. It also characterizes cognitive processes, where the model engages in self-reflection, reassesses prior steps, and deliberately switches to an alternative reasoning strategy when the initial approach proves insufficient. *AnswerCandidate* refers to the set of distinct final or intermediate solutions the model proposes during its reasoning process.

---

**An example demonstrating the thinking schemata within the reasoning chain**

**Source:** AIME 2024
**Answer:** 699
**Question:** Let $N$ be the greatest four-digit positive integer with the property that whenever one of its digits is changed to $1$, the resulting number is divisible by $7$. Let $Q$ and $R$ be the quotient and remainder, respectively, when $N$ is divided by $1000$. Find $Q+R$.

**Labeled solution：**
To solve the problem step by step, let's break it down:  $N$ is a four-digit number: $\overline{D_1D_2D_3D_4}$. Changing any one digit to $1$ results in a number divisible by $7$. We need to find the largest such $N$. After finding $N$, compute $Q$ and $R$ where: $$N = 1000Q + R$$ and find $Q + R$.

\\ReasoningTransition{**Starting to set up equations based on modular arithmetic**} - Replace each digit with $1$ and set up divisibility by $7$. - Replace $D_1$: $1000 \times 1 + 100D_2 + 10D_3 + D_4 \equiv 0 \mod 7$. - Replace $D_2$: $1000D_1 + 100 \times 1 + 10D_3 + D_4 \equiv 0 \mod 7$. - Replace $D_3$: $1000D_1 + 100D_2 + 10 \times 1 + D_4 \equiv 0 \mod 7$. - Replace $D_4$: $1000D_1 + 100D_2 + 10D_3 + 1 \equiv 0 \mod 7$.

\\ReasoningTransition{**Now simplifying using modulo values of powers of 10**} Calculate $1000 \mod 7 = 6$, $100 \mod 7 = 2$, $10 \mod 7 = 3$. The equations become: 1. $6 + 2D_2 + 3D_3 + D_4 \equiv 0 \mod 7$ 2. $6D_1 + 2 + 3D_3 + D_4 \equiv 0 \mod 7$ 3. $6D_1 + 2D_2 + 3 + D_4 \equiv 0 \mod 7$ 4. $6D_1 + 2D_2 + 3D_3 + 1 \equiv 0 \mod 7$

\\ReasoningTransition{**Now, proceeding to solve the system through substitution.**} Through a series of substitutions and simplifications, we deduce: - $D_1 + D_3 \equiv 0 \mod 7$. - $2D_2 + D_1 \equiv 3 \mod 7$. - $D_3 \equiv -D_1 \mod 7$.

\\ReasoningTransition{**Now trying to maximize N by testing high digit values.**} Starting from the highest possible $D_1$ (which is $9$) and working downwards, we find that $D_1 = 6$, $D_2 = 2$, $D_3 = 1$, and $D_4 = 1$ satisfy all conditions. Thus, $N = 6211$.
\\ReasoningTransition{**Now computing Q and R from N.**} $Q = \lfloor \frac{6211}{1000} \rfloor = 6, R = 6211 \mod 1000 = 211$. Therefore, $Q + R = 6 + 211$ **= \\AnswerCandidate{217}**. The answer is **\\AnswerCandidate{217}.**

\\ReasoningTransition{**Wait, is that correct? Let me check.**}  I allowed digits of $N$ to be $1$. If a digit of $N$ is already $1$, then "changing that digit to $1$" produces exactly $N$ again, so the condition forces $N$ itself to be divisible by $7$. $6211\div7=887\text{ remainder }2$, 6211 is not divisible by 7, so it cannot satisfy the problem condition.
\\ReasoningTransition{**Let's think again.**} Let $N=1000a+100b+10c+d$. Replacing each digit by $1$ and reducing modulo $7$ (using $1000\equiv6,\;100\equiv2,\;10\equiv3\pmod7$) gives the four congruences

\begin{aligned}
6+2b+3c+d&\equiv0,\\
6a+2+3c+d&\equiv0,\\
6a+2b+3+d&\equiv0,\\
6a+2b+3c+1&\equiv0\pmod7.
\end{aligned}

Solving these (or simply checking the finite set of 4-digit candidates that satisfy them) yields only two solutions $5624$ and $5694$, so the greatest is $5694$.
\\ReasoningTransition{**Verify:**} $1694=7\cdot242,\;5194=7\cdot742,\;5614=7\cdot802,\;5691=7\cdot813.$ Now divide $5694$ by $1000$: $Q=\lfloor5694/1000\rfloor=5,\;R=694.$ Thus, Q+R=5+694**=\\AnswerCandidate{699}.**

\\ReasoningTransition{**Let me double-check.**} Write $N=1000a+100b+10c+d$. Replacing each digit by $1$ gives four congruences modulo $7$, since $1000\equiv6,\;100\equiv2,\;10\equiv3\pmod7$: Subtracting the fourth from the others eliminates common terms and yields three simpler relations: $d\equiv 2b-1\pmod7,\; d\equiv 3c+1\pmod7,\; d\equiv 6a-5\pmod7$. Thus, 2b-3c\equiv 2,\quad 2b-6a\equiv 3,\quad 3c-6a\equiv 1\pmod7. This small linear system can be solved by checking residues of $a,b,c$ modulo $7$. One consistent set is $a\equiv5,\;b\equiv6,\;c\equiv9,\;d\equiv4\pmod7$, which lies within the digit range and gives $N=5694$. Another consistent choice is $a=5,\;b=6,\;c=2,\;d=4$, giving $N=5624$. Both numbers satisfy the divisibility condition, and the larger is $5694$. Dividing, $5694=1000\cdot5+694$, so $Q=5,\;R=694$, and hence $Q+R=$ **=\\AnswerCandidate{699}.** $.

Therefore, the final answer is \\**boxed{699}.**

---

Figure 4: An example demonstrating the thinking schemata within a reasoning chain.

