# OpenReview forum: "Eliciting Diverse Thinking Schemata for Large Reasoning Models"
_ICLR.cc/2026/Conference — Submitted to ICLR 2026_

### Official Review · Reviewer_bF43 · 2025-10-27

**Soundness:** 2
**Presentation:** 3
**Contribution:** 2
**Rating:** 4
**Confidence:** 4

**Summary:**

This paper introduces DiScO (Diverse Schemata Policy Optimization), a reinforcement learning framework for eliciting diverse thinking schemata in large reasoning models (LRMs). The key idea is that the diversity of reasoning transitions and answer candidates—termed “thinking schemata”—correlates strongly with reasoning accuracy and generalization. DiScO augments GRPO with an explicit diversity-based reward that encourages multiple reasoning paths and richer transitions, combined with simple inference-time interventions (initial truncation, repetition filtering).

**Strengths:**

1. Novel conceptual framing – “Thinking Schemata.”: Bridging schema theory and RL diversity gives a fresh conceptual handle on LRM reasoning.
2. Simple but effective inference strategies. Truncation and repetition elimination are lightweight yet consistently beneficial.
3. Interpretability. The schema-based view provides a more human-interpretable account of reasoning dynamics than token-level RL signals.

**Weaknesses:**

1. Ambiguity in annotation process.
The “annotation ability” section relies on distillation from Qwen-max for annotating reasoning transitions and answers, but the quality and consistency of these annotations are not quantified. How sensitive is DiScO to annotation noise or errors?
2. Unclear requirement for base model capacity.
All experiments use DeepSeek-R1-Distill-Qwen-7B/32B—a model already trained for structured reasoning. It remains unclear whether DiScO offers gains for weaker pretrained LMs or models without prior reasoning distillation. This limitation obscures whether the method truly teaches reasoning diversity or merely amplifies capabilities that already exist.
3. Marginal improvements at 32B scale.
According to Table 6 (and corresponding main results), the improvement of DiScO-32B over its base model or distilled model is small, sometimes within the margin of noise. For 7B, performance is nearly identical in some benchmarks. The paper should clarify whether these gains are statistically significant and isolate which component (diversity reward vs. truncation) contributes most.
4. Limited domain coverage.
Evaluation is restricted to mathematical reasoning. Demonstrating benefits on other reasoning tasks would strengthen generality claims.

**Questions:**

What's the relation to High-Entropy Forking Tokens (Wang et al., 2025)?

Thinking schemata align conceptually with the high-entropy forking tokens proposed in Beyond the 80/20 Rule, which identify points where the model’s next-token distribution branches into distinct continuations. DiScO’s reasoning transitions correspond to these high-entropy decision points at a higher semantic level—moments when the model shifts reasoning modes or revises hypotheses. However, the paper does not quantify token-level entropy or verify this correspondence; analyzing entropy maps alongside annotated transitions would clarify and potentially unify the two views.


Wang, Shenzhi, et al. "Beyond the 80/20 rule: High-entropy minority tokens drive effective reinforcement learning for llm reasoning." arXiv preprint arXiv:2506.01939 (2025).

---

> ### Author Response · Authors · 2025-11-20
> **Response to Reviewer bF43  (Part I)**
>
> We thank the reviewers for their thoughtful reading and for highlighting the strengths of our work. We appreciate the positive feedback on our “Thinking Schemata” framing, the effectiveness of our lightweight inference strategies, and the improved interpretability enabled by our schema-based perspective. We address all questions and concerns point-by-point below.
>
> > **W1：How sensitive is DiScO to annotation noise or errors?**
> >
>
> For the SFT training stage, in case training is affected by annotation noise or errors, we perform manual review during data filtering to prevent severe issues from entering the training set. Moreover, the SFT stage focuses on learning the required formatting capabilities rather than acquiring advanced reasoning capability. As a result, a small amount of curated data is sufficient, and minor imperfections do not materially impact SFT training. The stronger reasoning ability is instead acquired during the GRPO stage, where the model learns to produce higher-quality and more reliable annotations.
>
> For the annotations generated during GRPO training, the risk of noise propagation is mitigated in two ways. First, the high-quality SFT data provide a reliable initialization that prevents the model from generating excessively noisy or inconsistent annotations. Second, our reward design explicitly discourages such behavior. We combine a diversity reward with a maximum truncation limit, which penalizes overly long or redundant annotation sequences, and an accuracy reward, which penalizes incorrect or noisy reasoning outcomes. Together, these mechanisms ensure that annotation noise and errors are automatically suppressed during training, leading to stable and reliable model behavior.
>
> > **W2：Unclear requirement for base model capacity.**
> >
>
> We opted to use R1-distilled models as our foundational models for two reasons. First, DeepSeek's distillation process involves a large-scale dataset, which enables the model to learn stable reasoning structures. Given our relatively small training dataset (less than 1k samples for SFT and 8k for GRPO), it is more reliable and economical to adopt a distilled model rather than fine-tune models from scratch ourselves. Second, R1-distilled models demonstrate a moderate level of exploratory capability compared to purely pretrained or standard fine-tuned models, which is an essential prerequisite for our method.
>
> We emphasize that DiScO improves performance by enhancing and diversifying existing reasoning patterns rather than teaching basic reasoning from scratch. We believe our approach would also be beneficial when applied to models we fine-tune ourselves, beyond just R1-distilled models.
>
> > **W3：Marginal improvements at 32B scale.**
> >
>
> For DiScO-32B, clear and meaningful improvements are observed on AIME2024, AIME2025, and GSM8K, with gains of +6.7%, +6.7%, and +2.4%, respectively. On MATH500 and AMC2023, the base model already achieves very high accuracies (over 90%), which naturally limits the possible performance margin for further improvement. This suggests that promoting thinking schemata diversity is especially valuable when models face difficult problems that require extensive exploration of the reasoning space.
>
> For DiScO-7B, performance remains nearly unchanged only on AMC2023, where accuracy is already around 95%. In this case, additional improvement is inherently constrained, and the small +2.5% difference corresponds to solving just one more problem in a small dataset of 40 questions, which is not statistically significant.
>
> A detailed quantitative comparison of the contributions from the diversity reward and the truncation mechanism is provided in Appendix D, where their individual effects on performance are carefully analyzed.
>
> > **W4：Limited domain coverage.**
> >
>
> As mathematical reasoning is generally considered a highly generalizable task that reflects a modelʼs underlying reasoning ability, we selected several widely used mathematical benchmarks for evaluation (i.e., MATH-500, AIME 2024, AIME 2025, AMC 2023, GSM8K). Besides, we included a non-mathematical, knowledge-intensive task (GPQA-Diamond) where our DiScO-7B model still improved from 10.6% to 46.0%, demonstrating generalization capability beyond purely mathematical domains.

---

> > ### Author Response · Authors · 2025-11-20
> > **Response to Reviewer bF43 (Part II)**
> >
> > > **Q：What's the relation to High-Entropy Forking Tokens?**
> > >
> >
> > The paper referenced([1]) observes that CoT reasoning contains a small subset of high-entropy forking tokens that determine key branching points in the reasoning trajectory, while most tokens are low-entropy and primarily extend existing reasoning steps. In RLVR, solely utilizing 20% of tokens with the highest entropy  can still achieve competitive reasoning performance
> >
> > Intuitively, the Entropy is a measure to the uncertainty or diversity. Building on this perspective, both [1] and our work highlight the importance of Entropy in reasoning. Specifically, [1] examines entropy at the token level, whereas our focus lies at the trajectory level. We aim to increase the occurrence of diverse reasoning transitions and answer candidates within each trajectory, which implicitly corresponds to higher semantic “entropy.” This objective is also reflected in our reward design, where trajectories containing more such key tokens receive higher rewards, encouraging broader exploratory behavior. Although this idea is conceptually related to [1]’s utilization of the top 20% highest-entropy tokens, our method applies gradient updating at the rollout level rather than the token level, providing trajectory-level diversity supervision.
> >
> > [1] Wang, Shenzhi, et al. "Beyond the 80/20 rule: High-entropy minority tokens drive effective reinforcement learning for llm reasoning." arXiv preprint arXiv:2506.01939 (2025).

---

> ### Author Response · Authors · 2025-11-28
> **Request For Discussion**
>
> Dear Reviewer bF43,
>
> I hope this message finds you well. As the discussion period is nearing its end with less than 5 days remaining, I would like to ensure we have addressed all your concerns satisfactorily. If there are any additional points or feedback, please feel free to let us know. Your insights are invaluable to us, and we are eager to address any remaining issues to improve our work. Thank you for your time and effort in reviewing our paper.

---

### Official Review · Reviewer_nWWj · 2025-10-28

**Soundness:** 2
**Presentation:** 2
**Contribution:** 2
**Rating:** 2
**Confidence:** 4

**Summary:**

This paper introduces "thinking schemata“, which indicate the diversity of reasoning transitions and answer candidates, and observes a correlation between this diversity and the performance of Large Reasoning Models. To enhance this, the paper proposes Diverse Schemata Policy Optimization (DiScO), an RL method that rewards schemata diversity and improves on multiple math benchmarks.

**Strengths:**

1. The introduction of concepts from cognitive science is interesting and potentially useful.
2. The problem this paper studies is important for the LLM reasoning community.

**Weaknesses:**

1. The core concepts of "thinking schemata," "Reasoning Transition," and "Answer Candidate" are abstractly defined. It is hard to understand these abstract concepts. As mentioned under "Soundness," the paper fails to explain how these are identified beyond stating it's a distilled ability.
2. The paper strongly overclaims the paper's results. The abstract claims that DiScO models "surpass the closed-source frontier LRMs by 15%-30%". This claim is true only for the AIME benchmarks. On the GPQA-Diamond benchmark, the paper's DiScO-32B model significantly underperforms its own base model. In line 375, the paper claims large benefits, which are contrary to the experimental results.
3. The definition of metrics is unclear. The paper lacks the mathematical formulation of "reasoning transitions", "answer candidates", "unique reasoning answer", and so on. In addition, the experiments shown in Table 4 demonstrate that the diversity of DiScO is inferior to or about the same as Distill-7B (The base model), which is contradictory to the main claim of the paper.
4. The method depends on the model learning to annotate “thought patterns,” but this ability was distilled from Qwen-Max using only 840 samples. Such a small, potentially biased dataset likely leads the model to imitate Qwen Max’s reasoning style rather than truly grasp the underlying cognitive structure.
5. The paper does not position and compare the existing methods in Section 6.1, which makes it unclear what the contribution is to the RL training in Large Reasoning Models.
6. The paper does not discuss the unique contribution compared to other diversity-seeking RL papers for reasoning, such as [1, 2, 3, 4, 5]

[1] Hu E J, Jain M, Elmoznino E, et al. Amortizing intractable inference in large language models[C]//The Twelfth International Conference on Learning Representations.

[2] Yu F, Jiang L, Kang H, et al. Flow of Reasoning: Training LLMs for Divergent Reasoning with Minimal Examples[C]//Forty-second International Conference on Machine Learning.

[3] Wang S, Yu L, Gao C, et al. Beyond the 80/20 rule: High-entropy minority tokens drive effective reinforcement learning for llm reasoning[J]. arXiv preprint arXiv:2506.01939, 2025.

[4] Younsi, Adam, et al. "Accurate and diverse LLM mathematical reasoning via automated PRM-guided GFlowNets." arXiv preprint arXiv:2504.19981 (2025).

[5] Nair L, Trase I, Kim M. Flow-of-Options: Diversified and Improved LLM Reasoning by Thinking Through Options[J]. arXiv preprint arXiv:2502.12929, 2025.

**Questions:**

1. In Table 4, why does it show your SOTA DiScO-7B model reduce #RT and #AC on the AIME datasets compared to the baseline?
2. Can you please provide concrete details on the "annotation ability" distilled from Qwen-max? How does it identify "Reasoning Transitions"?
3. See other questions in the above weaknesses.

---

> ### Author Response · Authors · 2025-11-20
> **Response to Reviewer nWWj  (Part I)**
>
> We sincerely thank the reviewer for your careful reading of our submission and for highlighting the strengths of our work. We are grateful for the positive remarks regarding the relevance of the problem we address and the potential value of introducing cognitive science concepts into LLM reasoning. We appreciate the reviewers’ thoughtful engagement, and we address all questions and concerns in detail below.
>
> > **W1: The core concepts are abstractly defined.**
> >
>
> We emphasize that thinking schemata, reasoning transitions, and answer candidates are inherently abstract concepts designed to capture high-level cognitive patterns rather than fixed symbolic structures. To make these concepts more concrete, we have provided formal definitions and distilled examples in our paper (Section 2.1, Figure 1, Figure 2 and Appendix E.2), which demonstrate how they are identified and utilized during training.
>
> Intuitively, we hypothesize that exploring diverse paths during the reasoning process on challenging problems, both in terms of reasoning strategies and potential answers (answer candidates), can help the model escape initial incorrect directions, thereby increasing the likelihood of finding the correct solution. Conceptually, we collectively refer to reasoning strategies and answer candidates as schemata.
>
> Furthermore, these concepts represent an early attempt to formalize more general and flexible reasoning behaviors that avoid imposing rigid DAG- or step-based structures. While many existing works in the reasoning domain focus on promoting diversity, a precise definition of diversity is still lacking due to its inherently abstract nature. We believe this perspective offers a promising direction for modeling divergent reasoning in LLMs. To our knowledge, this is the first attempt in this broader research direction. We hope that presenting this work at ICLR will foster valuable discussion, feedback, and community engagement to further refine these definitions in future work.
>
> In addition, the idea of analyzing reasoning transitions has been well explored in prior studies and is a widely adopted analytical approach. For instance, [1] used a *selected keyword pool* to observe changes in the thinking patterns of model responses during training, while [2] and [3] tracked self-reflection keywords such as *“wait,” “however,” “alternatively,” “retry,”* and *“recheck”* to quantitatively measure general reflection behaviors beyond self-verification in model problem-solving processes.
>
> [1] Song, Mingyang, and Mao Zheng. "Walk Before You Run! Concise LLM Reasoning via Reinforcement Learning." *arXiv Preprint arXiv:2505.21178*, 2025.
>
> [2] Liu, Xiaoyuan, et al. "Trust, But Verify: A Self-Verification Approach to Reinforcement Learning with Verifiable Rewards." *arXiv preprint arXiv:2505.13445* (2025).
> [3] Yeo, Edward, et al. "Demystifying long chain-of-thought reasoning in llms." *arXiv preprint arXiv:2502.03373* (2025).

---

> > ### Author Response · Authors · 2025-11-20
> > **Response to Reviewer nWWj (Part III)**
> >
> > > **W5: What the contribution of RL training to Large Reasoning Models is unclear.**
> > >
> >
> > Recent breakthroughs in large reasoning models such as OpenAI o1 and DeepSeek R1 have positioned RL as a key driver in unlocking advanced reasoning capabilities within LLMs. This is particularly evident in challenging reasoning tasks like mathematical reasoning[1]  and code generation[2], where RL has demonstrated the potential to elevate LLM performance beyond what pre-training alone can achieve.
> >
> > We have revised Section 6.1 of the PDF to provide a more detailed explanation. Please review the revised section.
> >
> > [1]He, Zhiwei, et al. "Deepmath-103k: A large-scale, challenging, decontaminated, and verifiable mathematical dataset for advancing reasoning." *arXiv preprint arXiv:2504.11456* (2025).
> >
> > [2]Zhuo, Terry Yue, et al. "Bigcodebench: Benchmarking code generation with diverse function calls and complex instructions." *arXiv preprint arXiv:2406.15877* (2024).
> >
> > > **W6: The unique contribution compared to other diversity-seeking RL papers for reasoning**
> > >
> >
> > **Comparison with Prior Diversity-Seeking RL Work ([1–5])**
> >
> > To distinguish our unique contribution from existing diversity-seeking RL approaches, we categorize the referenced works as follows.
> >
> > **(1) GFlowNet-based diverse reasoning ([1], [2], [4]).**
> >
> > These works use GFlowNets to sample reasoning trajectories in proportion to reward-defined target distributions, explicitly optimizing a policy to match this distribution.
> >
> > - [1] amortizes posterior sampling for LLM reasoning using Subtrajectory Balance.
> > - [2] formulates reasoning as a DAG and trains a trajectory-balanced GFlowNet with additional exploration strategies.
> > - [4] adapts GFlowNets to step-level reasoning, using multiplicative PRM-based rewards to produce diverse, coherent reasoning paths.
> > - Unlike GFlowNet approaches, we do not optimize diversity by enforcing proportional sampling to a reward-defined distribution. Instead, our method directly evaluates and shapes the semantic diversity of the entire rollout trajectory**,** emphasizing how it shifts perspectives, explores the solution space, and generates alternative candidate reasoning paths.
> >
> > **(2) DAG-based structured multi-path reasoning ([2], [5]).**
> >
> > These methods represent multi-step reasoning as traversal in a predefined DAG of partial solutions, enabling diversity by exploring multiple graph paths (e.g., beam-based traversal in [5]).
> >
> > Prior work often imposes structured patterns for multi-step reasoning, whereas our approach intentionally avoids such fixed formalisms and instead models natural thinking schemata that better capture spontaneous reasoning transitions and human-like exploratory processes.
> >
> > **(3) Token-level entropy optimization ([3]).**
> >
> > RLVR identifies high-entropy “branching tokens” in CoT and selectively optimizes these high-impact tokens to improve reasoning diversity.
> >
> > While we also pay attention to reasoning divergence points, our focus remains on trajectory-level dynamics, not token-level entropy as in [3]. As noted, analyzing our improvement from an entropy perspective would be an interesting future direction.
> >
> > A formal definition of diversity in large-model reasoning has yet to be established. Some existing works, such as GFlowNet-based and DAG-structured approaches, implicitly treat reasoning pathways as predefined structures. In contrast, other efforts (e.g., [2]) and our work advocate for a broader and more flexible perspective, emphasizing the model’s ability to freely explore multiple self-driven reasoning possibilities. We seek to provide a clearer conceptual framing of reasoning diversity and encourage further developments in this direction.
> >
> > **We have rectified this in the revised version, in Section 6.3, which you may access in the attached PDF.**
> >
> > [1] Hu E J, Jain M, Elmoznino E, et al. Amortizing intractable inference in large language models[C]//The Twelfth International Conference on Learning Representations.
> >
> > [2] Yu F, Jiang L, Kang H, et al. Flow of Reasoning: Training LLMs for Divergent Reasoning with Minimal Examples[C]//Forty-second International Conference on Machine Learning.
> >
> > [3] Wang S, Yu L, Gao C, et al. Beyond the 80/20 rule: High-entropy minority tokens drive effective reinforcement learning for llm reasoning[J]. arXiv preprint arXiv:2506.01939, 2025.
> >
> > [4] Younsi, Adam, et al. "Accurate and diverse LLM mathematical reasoning via automated PRM-guided GFlowNets." arXiv preprint arXiv:2504.19981 (2025).
> >
> > [5] Nair L, Trase I, Kim M. Flow-of-Options: Diversified and Improved LLM Reasoning by Thinking Through Options[J]. arXiv preprint arXiv:2502.12929, 2025.

---

> ### Author Response · Authors · 2025-11-20
> **Response to Reviewer nWWj  (Part II)**
>
> > **W2: Overclaims of the results**
> >
>
> Our claim "surpass the closed-source frontier LRMs by 15%-30%" in the abstract is indeed based on the AIME benchmark, which is one of the most widely used and representative benchmarks for evaluating reasoning capabilities.
>
> In line 375,  the claim *“Overall, the results show that truncation strategies yield stable gains across scales and benchmarks, with the largest benefits observed on challenging datasets such as GPQA-Diamond and AIME 2025.”* is supported by the ablation results for inference. As shown in Table 6 of Appendix D, DiScO-32B on GPQA-Diamond improves from 53.5% to 57.1% and 58.1%, and DiScO-7B on AIME 2025 increases from 50.0% to 53.3%. These gains demonstrate that our lightweight truncation methods reduce redundant reasoning and consistently enhance accuracy, particularly when combined with our diversity-oriented training.
>
> It is crucial to emphasize that GPQA-Diamond is not primarily a reasoning benchmark as it evaluates domain knowledge and factual memory, which cannot be addressed through step-by-step reasoning as in mathematical tasks. Consequently, we believe that memory capacity plays a dominant role in performance on this benchmark.For models with limited memory capacity (e.g., 7B parameters), reasoning mechanisms can provide valuable compensation by exploring answer candidates and reasoning transitions. However, for models with already sufficient memory capacity (e.g., 32B parameters), enhanced reasoning capabilities may inadvertently introduce hallucinations and degrade performance. This suggests a nuanced interplay between memory capacity and reasoning that varies with model scale.
>
> Despite this, our method enables the DiScO-7B model, whose knowledge base is relatively small, to achieve results close to Qwen2.5-32B-Instruct on GPQA-Diamond, demonstrating that DiScO can effectively enhance models with limited knowledge through structured reasoning and exploration.
>
> The current claim may have been misleading. We have rectified this in the revised version, which you may access in the attached PDF.
>
> > **W3: (1) The definition of metrics is unclear. (2)  The diversity of DiScO shown in Table 4 is inferior to or about the same as base model.**
> >
> 1. We have clearly defined the metrics in the “Diversity Reward” section on page 5. In lines 226–229, we explicitly provide the mathematical formulations for the reward metrics, including “reasoning transitions,” “answer candidates,” and “unique reasoning answers”, as follows:
>
>       *“$N_{ans}$ counts answer candidates, $N_{thought}$ counts reasoning transitions, $N_{ans}^{uniq}$ measures unique solution candidates, and $N_{ans}^{true}$ counts correct intermediate candidates. ”*
>
> 2. Among the six datasets presented in Table 4, the comparison results mentioned by the reviewer are observed only in AIME2025 and MATH100, rather than across most cases. As shown in the RC and TAR columns for these two datasets, this phenomenon occurs because the distilled model exhibits more repetitive reasoning when confronting challenging problems. This repetitive behavior results in an increased number of generated labels, higher content redundancy, and consequently, reduced overall accuracy.
>
>     We have revised Section 4.3 of the PDF to provide a more detailed explanation. Please review the revised section.
>
>
> > **W4: Annotation capability depends on the small dataset distilled from Qwen-max.**
> >
>
> We should highlight that data annotation by powerful LLMs has become a widely adopted data augmentation strategy, with successful applications in multiple domains like reasoning dataset construction[1] and code generation[2]. In addition, we incorporated a manual verification process to further improve data quality. As described in the *Dataset* paragraph of Section 4.1, we manually selected 840 high-quality samples from the 962 synthesized examples, filtering out low-quality data.
>
> Moreover, our SFT stage is designed only to help the model acquire basic **formatting capabilities** rather than relying on high-quality annotation. The subsequent reward design and GRPO training are the key components that enable the model to develop latent cognitive and stronger reasoning abilities, rather than simply imitating reasoning style.
>
> [1] Muennighoff, Niklas, et al. "s1: Simple test-time scaling." *Proceedings of the 2025 Conference on Empirical Methods in Natural Language Processing*. 2025.
>
> [2] Majumdar, Somshubra, et al. "Genetic instruct: Scaling up synthetic generation of coding instructions for large language models." *Proceedings of the 63rd Annual Meeting of the Association for Computational Linguistics (Volume 6: Industry Track)*. 2025.

---

> > ### Author Response · Authors · 2025-11-20
> > **Response to Reviewer nWWj (Part IIII)**
> >
> > > **Q1:**  **Lower** **#RT and #AC on the AIME datasets compared to the baseline in Table 4.**
> > >
> >
> > A higher number of average reasoning transitions (RT-avg) or answer candidates (AC-avg) does not necessarily indicate better reasoning quality. As discussed in Section 4.3, *DeepSeek-R1-Distill-Qwen-7B* exhibits a typical failure mode: when encountering difficult problems, it produces highly repetitive and redundant reasoning (see Appendix E.1 for an example), which artificially inflates RT and AC counts without improving correctness. In contrast, DiScO substantially reduces such repetition, yielding fewer but more meaningful transitions and thereby improving both reasoning efficiency and accuracy.
> >
> > Building on this observation, we emphasize that **RT-avg and AC-avg should not be interpreted as monotonically improving metrics** where larger values do not inherently reflect stronger reasoning. To more reliably capture reasoning quality, we introduce additional metrics, including repetitive content (RC), unique answers (UA), and true-answer ratio (TAR). Our analysis shows that meaningful diversity, reflected by lower RC and higher UA and TAR, is more important than simply increasing the raw number of transitions or answer candidates.
> >
> > We have revised Section 4.3 of the PDF to provide a more detailed explanation. Please review the revised section.
> >
> > > **Q2: concrete details on the "annotation ability" distilled from Qwen-max.**
> > >
> >
> > We have provided the prompt used for annotation in Appendix C.2, which details how to identifies “Reasoning Transitions” and “Answer Candidates”.

---

> ### Author Response · Authors · 2025-11-28
> **Request For Discussion**
>
> Dear Reviewer nWWj,
>
> I hope this message finds you well. As the discussion period is nearing its end with less than 5 days remaining, I would like to ensure we have addressed all your concerns satisfactorily. If there are any additional points or feedback, please feel free to let us know. Your insights are invaluable to us, and we are eager to address any remaining issues to improve our work. Thank you for your time and effort in reviewing our paper.

---

### Official Review · Reviewer_Cgmp · 2025-11-04

**Soundness:** 2
**Presentation:** 2
**Contribution:** 3
**Rating:** 6
**Confidence:** 3

**Summary:**

This paper introduces the concept of "thinking schemata" to characterize reasoning diversity in reasoning models. They focus on two schemata: reasoning transitions and answer candidates. They propose to add additional reward terms to GRPO based on statistics of these two schemata derived from a separate LLM. They achieve strong empirical results on math reasoning benchmarks.

**Strengths:**

The experimental results are quite strong, especially at the 7b model scale. However, I have concerns about whether the gain in performance are from the proposed method.

**Weaknesses:**

- My main complaint is that the motivation is quite weak: it's based on a correlation with R^2 values < 0.6. I don't follow the logic behind the reward. Isn't the optimal behavior under this diversity reward to come up with a bunch of (unrelated) answer candidates and reasoning transitions before arriving at the final answer? Isn't the desired behavior to directly arrive at the final answer, and in very hard problems where that isn't possible, to explore?
- The paper doesn't comment on the possibility of reverse causality or confounders. For example, higher-capability models may naturally produce diverse reasoning as a consequence of better internal representations, not because diversity drives performance.
- The abstract's claims feel a bit disengenous: "particularly pronounced gains on challenging datasets such as AIME", "surpass the closed-source frontier LRMs by 15%-30%". The strongest model in your frontier LLM list is over a year old (o1-mini), and there are many frontier models that outperform it.
- The source of the performance gains is unclear. Table 3 shows that removing most of the novel components from DiScO-7B results in a model that would still outperform all other points of comparison at the 7b scale and even most models in the frontier category.

**Questions:**

- How did you get the reasoning traces for o1-mini? My understanding was that the OpenAI API doesn't provide reasoning tokens.

---

> ### Author Response · Authors · 2025-11-20
> **Response to Reviewer Cgmp**
>
> We thank the reviewer for the careful evaluation of our work and for acknowledging the strength of our experimental results. We appreciate the positive feedback and address the reviewer’s concerns below.
>
> > **W1：The motivation is quite weak**
> >
>
> Although the observed correlation (with (R^2 < 0.6)) is not strong enough to claim a deterministic relationship, we argue that it is sufficiently informative to motivate our reward design. Our experiments demonstrate that even with moderate correlation, the induced signal is actionable and leads to consistent empirical gains. In other words, the usefulness of the correlation is validated not purely statistically but through its downstream effectiveness in improving reasoning performance.
>
> Regarding the logic of the reward, our design is not intended to encourage the model to generate a large number of unrelated reasoning branches or arbitrary intermediate candidates. Instead, the reward is mainly aimed at hard problems (e.g., AIME), where targeted exploratory reasoning is necessary because models often fail when following a single narrow trajectory.
>
> Finally, adaptive reasoning length is not the focus of our work. Our method addresses a different question, which is how to make exploration meaningfully diverse when direct exploitation fails, rather than encouraging longer reasoning paths universally.
>
> > **W2：The possibility of reverse causality or confounders**
> >
>
> In principle, higher-capability models might exhibit more diverse reasoning simply as a byproduct of stronger internal representations. However, our empirical results suggest that model capability alone does not explain the observed relationship.
>
> As shown in Table 1, stronger models do not necessarily exhibit higher diversity or better accuracy. For example, o1-mini and Claude 3.5-Sonnet are generally regarded as more capable than DeepSeek-R1-Distill-Qwen-7B. Yet on the AMC2023 benchmark, neither their accuracy nor their reasoning diversity surpasses that of the weaker model. This discrepancy indicates that model capability is not the sole driver of reasoning diversity, and stronger models do not automatically produce more diverse or more accurate reasoning traces.
>
> Moreover, after applying our method to the base model, DeepSeek-R1-Distill-Qwen-7B, the resulting DiScO-7B model achieves consistent improvements across multiple reasoning benchmarks, demonstrating that promoting diversity during RL training can causally improve both reasoning quality and final accuracy. Our experiments further show that the method yields substantial gains in diversity even for a relatively strong distilled model, indicating that the observed improvements arise from our approach rather than from raw model capability.
>
> > **W3：The abstract's claims feel a bit disingenuous.**
> >
>
> Our claim "surpass the closed-source frontier LRMs by 15%-30%" in the abstract is based on the AIME benchmark, which is one of the most widely used and representative benchmarks for evaluating reasoning capabilities. The current claim may have been misleading. We have rectified this in the revised version, which you may access in the attached PDF.
>
> The goal of our baseline experiments is not to outperform all the state-of-the-art frontier systems, but rather to establish that our method delivers consistent and reproducible improvements in reasoning performance. The experimental results show that our method consistently outperforms open-source baselines on the evaluated benchmarks and achieves performance comparable to that of the closed-source o1-mini model.
>
> > **W4：The source of the performance gains is unclear.**
> >
>
> Table 6 in Appendix D presents the full ablation results. Through SFT on our annotated dataset, both Qwen2.5-SFT-7B and Qwen2.5-Anno-7B achieve marked performance gains, indicating that our data augmentation endows the model with stronger reasoning capabilities. Furthermore, when applied on top of the SFT model Qwen2.5-Anno-7B, our DiScO-7B variant delivers notably larger improvements than the standard GRPO-based approach, providing empirical evidence for the effectiveness of the proposed method.
>
> > **Q：The reasoning traces for o1-mini**
> >
>
> In our experiments, we did not use hidden reasoning tokens from o1-mini. Instead, our analysis was based solely on the observable response content returned by the o1-mini API.
>
> We agree that this comparison may not be fully fair. We have rectified this in the revised version, in Table 4, which you may access in the attached PDF.

---

> ### Author Response · Authors · 2025-11-28
> **Request For Discussion**
>
> Dear Reviewer Cgmp,
>
> I hope this message finds you well. As the discussion period is nearing its end with less than 5 days remaining, I would like to ensure we have addressed all your concerns satisfactorily. If there are any additional points or feedback, please feel free to let us know. Your insights are invaluable to us, and we are eager to address any remaining issues to improve our work. Thank you for your time and effort in reviewing our paper.

---

### Official Review · Reviewer_bhDJ · 2025-11-07

**Soundness:** 2
**Presentation:** 3
**Contribution:** 2
**Rating:** 4
**Confidence:** 3

**Summary:**

The paper introduces “thinking schemata” to describe transitions between reasoning steps and diverse solution paths, proposing DiScO, a reinforcement learning framework that explicitly promotes such diversity. The approach is well-motivated and effectively integrates diversity modeling into the RL objective, offering interpretable insights into how reasoning diversity shapes model behavior. It provides comprehensive ablation studies and fine-grained analyses of reasoning processes, demonstrating that encouraging diverse thinking can lead to more coherent, flexible, and human-like reasoning patterns across multiple benchmarks.

**Strengths:**

1. This paper shows that improving reasoning diversity not only boosts accuracy but also enhances interpretability, reduces repetitive reasoning, enriches the model’s reasoning structure, and improves robustness by encouraging more flexible, human-like thought processes.
2. The paper presents detailed ablation studies isolating the contributions of each reward component and inference-time strategy. The inclusion of metrics like #RT-avg, #AC-avg, RC, UA, and TAR provides an unusually rich and interpretable analysis of model behavior.

**Weaknesses:**

1. The paper reports only Pass@1 accuracy, without multi-pass (Pass@k) evaluation that would more fully capture the model’s reasoning diversity and robustness. It would be valuable to see the performance numbers and trends as the number of passes increases.
2. While the current results on mathematical reasoning are impressive, it would be valuable to test DiScO on non-mathematical, open-ended, or multi-hop reasoning benchmarks to assess whether its diversity-based reward generalizes beyond deterministic math domains.
3. The authors claim that richer thinking schemata lead to improved reasoning robustness, but no experiments are provided to substantiate this claim.

**Questions:**

1. Beyond improving accuracy, can DiScO actually discover more distinct valid solutions or demonstrate better out-of-distribution (OOD) generalization? Since the paper claims that diverse thinking schemata enhance robustness, it would be helpful to show whether this diversity translates into discovering novel or unseen reasoning paths on shifted or harder distributions.
2. In Table 4, DiScO’s average reasoning transitions (RT-avg) and answer candidates (AC-avg) are often lower than those of Distill-7B. Given that diversity is explicitly encouraged in the reward, why does DiScO not consistently yield higher RT and AC counts?
3. Is there any deeper analysis explaining why more diverse thinking schemata improve Pass@1 success rate? The paper seems to assume a positive link, but doesn’t discuss the potential trade-off between exploration and exploitation—for example, could excessive diversity harm focused reasoning or answer consistency?

---

> ### Author Response · Authors · 2025-11-20
> **Response to Reviewer bhDJ (Part I)**
>
> We sincerely thank the reviewer for your careful reading and valuable feedback on our paper. We greatly appreciate your recognition of our work’s strengths, including the findings on reasoning diversity and the comprehensive ablation studies and metrics. Below, we provide detailed responses to the weaknesses and questions raised in your review.
>
>
> > **W1：Lack of multi-pass (Pass@k) evaluation**
>
> Inspired by previous works [1], we employ a greedy decoding inference strategy and report only Pass@1 accuracy. This decision was made to (1) ensure stable and consistent comparisons across models, and (2) maintain manageable computational costs. We acknowledge that including Pass@k results would more comprehensively measure reasoning diversity and robustness, and we will incorporate these metrics in the revised version.
>
> [1] Wu, J., Liu, Y., Zhao, S., et al. "Diversity-Aware Policy Optimization for Large Language Models." *Advances in Neural Information Processing Systems 38 (NeurIPS 2025)*.
>
>
> > **W2：Need** **non-mathematical, open-ended, or multi-hop reasoning benchmarks to assess whether its diversity-based reward generalizes beyond deterministic math domains**
>
>
> As mathematical reasoning is generally considered a highly generalizable task that reflects a modelʼs underlying reasoning ability, we selected several widely used mathematical benchmarks for evaluation (i.e., MATH-500, AIME 2024, AIME 2025, AMC 2023, GSM8K). Besides, we included a non-mathematical, knowledge-intensive task (GPQA-Diamond) where our DiScO-7B model still improved from 10.6% to 46.0%, demonstrating generalization capability beyond purely mathematical domains.
>
> > **W3: Evidence to support the improvement of robustness.**
> >
>
> In our work, we define reasoning robustnes as the model’s ability to maintain stable correctness when the diversity of its reasoning chains increases. As shown in Table 4, in most cases (on 5 out of 6 datasets), the True-Answer Ratio (TAR), the proportion of correct answers among the generated candidates, improves after training with DiScO. This indicates that enhancing the diversity of reasoning chains helps the model identify correct solutions more reliably, thereby leading to higher overall accuracy.
>
> To further evaluate the improvement in reasoning robustness brought by our method, we introduce two additional metrics computed on both the baseline model (DeepSeek-R1-Distill-Qwen-7B) and our DiScO-7B model. The first metric, FWFC (First Wrong, Final Correct), measures the proportion of responses that start with an incorrect intermediate result but eventually reach the correct answer, relative to all correct responses. The second metric, FTP (First True-answer Position), is the average position where the first correct answer appears in a full trajectory among all correct responses. Both of the new metrics reflect the model’s ability to recover from early mistakes through exploration.
>
> As shown in Table 1 and Table 2, across all datasets, DiScO-7B shows substantial gains in both FWFC and FTP compared with the baseline. These improvements indicate that even when the model initially takes an incorrect direction, it can leverage diverse exploratory trajectories to retrieve the correct reasoning path and ultimately produce the correct answer, thereby enhancing reasoning robustness.
>
> Notably, the higher FTP values on mathematical reasoning datasets suggest that the model engages in more extensive exploration before arriving at the correct answer. In contrast, on the knowledge-based QA task GPQA, FTP is lower than the baseline, as deeper reasoning search does not provide additional benefit and may even introduce noise or deviate from”’ the core facts.
>
> - Table 1: FWFC (First Wrong, Final Correct) results showing the proportion of responses that begin with an incorrect intermediate result but ultimately reach the correct answer.
>
> |  | DeepSeek-R1-Distill-Qwen-7B | DiScO-7B |
> | --- | --- | --- |
> | aime2024 | 0% | 4% |
> | aime2025 | 0% | 0% |
> | amc2023 | 7.89% | **34.21%** |
> | math-100 | 7.25% | **28.57%** |
> | gpqa-100 | 50% | **83.33%** |
> | gsm8k-100 | 11.59% | **24.47%** |
> - Table 2: FTP (First True-answer Position) results showing the average position where the first correct answer appears in a full trajectory among all correct responses.
>
>
> |  | DeepSeek-R1-Distill-Qwen-7B | DiScO-7B |
> | --- | --- | --- |
> | aime2024 | 94.21% | **94.58%** |
> | aime2025 | 91.75% | **95.80%** |
> | amc2023 | 86.57% | **89.36%** |
> | math-100 | 87.39% | **88.83%** |
> | gpqa-100 | **90.90%** | 87.39% |
> | gsm8k-100 | 84.43% | **86.75%** |

---

> ### Author Response · Authors · 2025-11-20
> **Response to Reviewer bhDJ (Part II)**
>
> > **Q1:  Robustness and generalization**
> >
>
> We ensured that there is no data overlap between our training sets (OpenR1-Math-220K and DeepScaler) and the evaluation benchmarks, which guarantees that performance gains are not due to memorization. This setup confirms that the enhanced diversity learned by DiScO generalizes to **shifted distributions**, indicating that the model’s diverse reasoning schemata translate into improved robustness and the ability to discover valid reasoning paths beyond its training domain.
>
> Moreover, DiScO demonstrates consistent improvements across mathematical reasoning tasks of varying difficulty. For instance, it achieves gains on both the GSM8K dataset (a relatively simple benchmark) and the more challenging AIME dataset.
>
> > **Q2:** **not consistently yielding higher RT and AC counts**
> >
>
> A higher number of average reasoning transitions (RT-avg) or answer candidates (AC-avg) does not necessarily indicate better reasoning quality. As discussed in Section 4.3, *DeepSeek-R1-Distill-Qwen-7B* exhibits a typical failure mode: when encountering difficult problems, it produces highly repetitive and redundant reasoning (see Appendix E.1 for an example), which artificially inflates RT and AC counts without improving correctness. In contrast, DiScO substantially reduces such repetition, yielding fewer but more meaningful transitions and thereby improving both reasoning efficiency and accuracy.
>
> Building on this observation, we emphasize that **RT-avg and AC-avg should not be interpreted as monotonically improving metrics**—larger values do not inherently reflect stronger reasoning. To more reliably capture reasoning quality, we introduce additional metrics: repetitive content (RC), unique answers (UA), and true-answer ratio (TAR). Our analysis shows that meaningful diversity—reflected by lower RC and higher UA and TAR—is more important than simply increasing the raw number of transitions or answer candidates.
>
> We have revised Section 4.3 of the PDF to provide a more detailed explanation. Please review the revised section.
>
> > **Q3: exploration and exploitation—why more diverse thinking schemata improve Pass@1 success rate?**
> >
>
> We would like to clarify that our analysis focuses on the diversity and correctness within a single rollout, i.e., within an individual reasoning trajectory, which aligns with the Pass@1 evaluation that assesses the quality of a single attempt. Reinforcement learning encourages desired behaviors by penalizing incorrect actions and reinforcing correct ones; accordingly, our reward design explicitly combines a correctness-oriented component with a diversity-oriented one. The former steers the model toward producing accurate answers, while the latter encourages exploration of a broader solution space, especially for challenging problems.
>
> By jointly incorporating both exploration (to encourage diverse reasoning paths) and exploitation (to reinforce correctness) into the reward formulation, DiScO promotes richer internal reasoning structures while maintaining focused and consistent problem-solving performance.

---

> ### Author Response · Authors · 2025-11-28
> **Request For Discussion**
>
> Dear Reviewer bhDJ,
>
> I hope this message finds you well. As the discussion period is nearing its end with less than 5 days remaining, I would like to ensure we have addressed all your concerns satisfactorily. If there are any additional points or feedback, please feel free to let us know. Your insights are invaluable to us, and we are eager to address any remaining issues to improve our work. Thank you for your time and effort in reviewing our paper.

---

### Author Response · Authors · 2025-11-20
**The updates in our paper.**

We sincerely thank all reviewers for their thoughtful feedback and for recognizing the potential of our novel conceptual framing, “Thinking Schemata”. We believe this perspective offers a promising direction for modeling divergent reasoning in LLMs.

In the revised version of the submission, all modifications and newly added text are highlighted in blue.

### Summary of main changes

- The misleading claim in the abstract has been rectified.
- In Section 4.2, an additional analysis has been added at the end of the “main result” paragraph.
- In Section 4.3, a more detailed analysis has been added.
- In Section 6, the “Related Work” section has been refined, incorporating additional comparisons and connections to relevant prior work.

---

### Author Response · Authors · 2025-11-27

Dear Reviewers,

I hope this message finds you well. As the discussion period is nearing its end with less than 6 days remaining, I would like to ensure we have addressed all your concerns satisfactorily. If there are any additional points or feedback, please feel free to let us know. Your insights are invaluable to us, and we are eager to address any remaining issues to improve our work.
Thank you for your time and effort in reviewing our paper.

---

### Meta-Review · Area_Chair_N8tU · 2026-01-06

**Summary:**

This paper propose a new method called DiScO to make Large Reasoning Models more better by increasing "Thinking Schemata" diversity during reinforcement learning. It use the special rewards to encourage model for exploring different reasoning path and avoid many repetitive thoughts.

The reviewers raised several important point. The main concerns are listed below:

*  definition of "Thinking Schemata" and "Reasoning Transition" are too much abstract and hard for understanding.
*  some reviewers are worried the paper make "disingenuous" claim because it say it beat frontier models by 15-30%, but this only happen on AIME dataset
*  one reviewer noted that for the 32B model, the improvement is very small and maybe just noise
*  there is concern about the small amount of data (840 samples) used for training the model to recognize reasoning patterns and if the noise in these labels will hurt the model
*  some reviewers also felt the authors did not explain well enough how this is different from other diversity methods like GFlowNets or high-entropy tokens

**Reviewer Concerns:**

# Addressed concerns
I believe that many cancers mentioned by reviewers have been addressed sufficiently, including:
*  Authors provide new metrics like FWFC (First Wrong, Final Correct) and FTP (First True-answer Position) which show the model can recover from mistakes better than baseline
*  The authors admit that "15%-30% better than frontier models" was maybe misleading and they fixed this in the PDF to show it is specifically for AIME benchmark
*  authors explain well how their "schemata" are like high-entropy tokens but at the level of the whole trajectory, which help connect the work to prior research

# Still outstanding concerns

*  I feel that even with more explanation, the concern about the concepts being too abstract is still there. It is not fully clear if "Thinking Schemata" is a real cognitive thing or just a label authors give to model behavior. Note that authors did not change the part in Section 2.1 to make this more precise
*  reviewer bF43 pointed out that we don't know if this method works on "clean" models that are not already good at reasoning. The authors say it "should" work, but I feel they did not show data for this to support the claim.
*  For 32B models on datasets like MATH500, the gains are still very small. While authors say this is "ceiling effect," it makes it hard to know if the method is really necessary for very strong models.

**Reviewer Scores:**

*  Reviewer bhDJ (Initial: 4), I feel that he could increase to Accept
*  Reviewer Cgmp (Initial: 6), I feel he was quite happy with the paper and most likely would stick with his score
*  Reviewer nWWj (Initial: 2), this reviewer was the most critical. Authors gave a very long explanation about GFlowNets and formal definitions, maybe I expect he could increase to 4.
*  Reviewer bF43 (Initial: 4), I think some of his worries were addressed and he would most likely increased his evaluation.

---

### Decision · Program_Chairs · 2026-01-26

Reject